

# From combinatorial maps to correlation functions in loop models

**Linnea Grans-Samuelsson[1,⋆,¶], Jesper Lykke Jacobsen[1,2,3†],**
**Rongvoram Nivesvivat[1‡ ‖] and Hubert Saleur[1,4§]**

**1** Institut de physique théorique, CEA, CNRS, Université Paris-Saclay
**2** Laboratoire de Physique de l'École Normale Supérieure, ENS,
Université PSL, CNRS, Sorbonne Université, Université de Paris
**3** Sorbonne Université, École Normale Supérieure, CNRS, Laboratoire de Physique (LPENS)
**4** Department of Physics and Astronomy, University of Southern California, Los Angeles

⋆ linneag@microsoft.com , † jesper.jacobsen@ens.fr ,
‡ rongvoram.n@outlook.com , ◦ sylvain.ribault@ipht.fr , § hubert.saleur@ipht.fr

## Abstract

In two-dimensional statistical physics, correlation functions of the $O(N)$ and Potts models may be written as sums over configurations of non-intersecting loops. We define sums associated to a large class of combinatorial maps (also known as ribbon graphs). We allow disconnected maps, but not maps that include monogons. Given a map with $n$ vertices, we obtain a function of the moduli of the corresponding punctured Riemann surface. Due to the map's combinatorial (rather than topological) nature, that function is single-valued, and we call it an $n$-point correlation function. We conjecture that in the critical limit, such functions form a basis of solutions of certain conformal bootstrap equations. They include all correlation functions of the $O(N)$ and Potts models, and correlation functions that do not belong to any known model. We test the conjecture by counting solutions of crossing symmetry for four-point functions on the sphere.

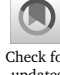

## Contents

¶ Now at: Microsoft Station Q, Santa Barbara, California 93106-6105 USA.
‖ Now at: Yau Mathematical Sciences Center, Tsinghua University, Beijing, 100084 China.

# 1 Introduction

## 1.1 Global symmetry in the $O(N)$ and Potts models

The $O(N)$ model and the $Q$-state Potts model are generalizations of the Ising model. In addition, they can be used for describing systems such as polymers, percolation or random walks. Both models can be defined on lattices, and have critical limits where they become conformal field theories on continuous spaces.

In addition to conformal symmetry, these field theories also enjoy the global symmetries of the original models, which are described by the orthogonal group $O(N)$ and the symmetric group $S_Q$ respectively. It is therefore natural to characterize the fields by their transformation properties under conformal symmetry and global symmetry. The action of global symmetry is described by finite-dimensional representations of the symmetry group. Each representation describes the behaviour of infinitely many primary fields. For example, in the case of $O(N)$,

the two simplest irreducible representations are the singlet and vector representations, and the $O(N)$ model has infinitely many primary fields that are $O(N)$ singlets or vectors.

In two dimensions, the action of the global and conformal symmetries have recently been determined [1]. A striking feature of the resulting spectra is the presence of degeneracies: two fields that transform in different irreducible representations of the symmetry group, or even two fields that belong to different models, can have the same conformal dimension. Let us illustrate this by displaying the numbers of primary fields for a few of the lowest conformal dimensions that appear in these models:

$$
\begin{array}{c|c|c}
\text{Dimension} & O(N) \text{ model} & \text{Potts model} \\
\hline
\delta_{\Delta_{(0,\frac{1}{2})}} & - & 1 \\
\delta_{(\frac{1}{2},0)} & 1 & - \\
\delta_{(1,0)} & 1 & - \\
\delta_{(\frac{3}{2},0)} & 2 & - \\
\delta_{(2,0)} & 5 & 1 \\
\delta_{(\frac{5}{2},0)} & 11 & - \\
\delta_{(3,0)} & 38 & 2
\end{array}
\tag{1}
$$

Our admittedly unconventional notations for conformal dimensions are explained in Table (60). By the number of primary fields we really mean the number of irreducible representations. For example, in the $O(N)$ model, the primary fields with the conformal dimension $\delta_{(\frac{5}{2},0)}$ transform in the representation [1](2.29i)

$$
\Lambda_{(\frac{5}{2},0)} = [5] + [32] + 2[311] + [221] + [11111] + [3] + 2[21] + [111] + [1], \tag{2}
$$

which is a sum of 9 different irreducible representations of $O(N)$ (written as integer partitions), including two representations [311] and [21] that come with nontrivial multiplicities.

The appearance of degeneracies in the two-dimensional $O(N)$ and Potts models can be understood in terms of the diagram algebras that are Schur–Weyl dual to the groups $O(N)$ and $S_Q$ in the lattice spectra of the models [1,2]. The problem is that these algebras are not dynamical symmetries, i.e. they do not constrain correlation functions. For example, in the $O(n)$ model, the operator product of two fields with dimensions $\delta_{(\frac{1}{2},0)}$ and $\delta_{(1,1)}$ involve fields with the dimension $\delta_{(\frac{5}{2},0)}$ that transform in the representations [111] and [21], but not in the rest of $\Lambda_{(\frac{5}{2},0)}$ [3]. The diagram algebra that has $\Lambda_{(\frac{5}{2},0)}$ as an irreducible representation can therefore help us understand the spectrum, but not the operator product expansions.

## 1.2 Making a mess of correlation functions

The inadequacy of global symmetry for taming the Potts and $O(N)$ models, and the lack of a sharp distinction between the two models, become even clearer at the level of correlation functions.

Correlation functions must be invariant under global symmetry. In particular, a four-point function of the $O(N)$ model must behave as an $O(N)$-invariant four-tensor. For example, there is one primary field $V_{(1,0)}$ of dimension $\delta_{(1,0)}$, which belongs to the symmetric two-tensor representation [2]. Since $[2] \otimes [2] = [4] + [31] + [22] + [2] + [11] + []$ is a sum of 6 irreducible representations, there are 6 invariant four-tensors in $[2]^{\otimes 4}$, and therefore 6 four-point functions of the type $\langle V_{(1,0)} V_{(1,0)} V_{(1,0)} V_{(1,0)} \rangle$ in the $O(N)$ model.

On the other hand, correlation functions are subject to conformal bootstrap equations — in the case of four-point functions on the sphere, crossing symmetry equations. These equations

depend only on conformal dimensions, and know nothing about global symmetry. As explained in more detail in Section 4.1, we can numerically determine the dimensions of spaces of solutions of crossing symmetry equations. We can then try to identify the solutions with the four-point functions that are predicted by global symmetry in the $O(N)$ or Potts model.

In the cases of a few four-point functions, let us display the numbers of $O(N)$- and $S_Q$-invariant four-tensors, together with the dimension of the space of solutions of crossing symmetry ("Bootstrap"). Our four-point functions may belong to the $O(N)$ model, the Potts model, neither, or both. Accordingly, the spectrum that we assume in the crossing symmetry equations is that of the $O(N)$ model, the Potts model, or the union of the two spectra.

| Four-point function | Models | $O(N)$ | $S_Q$ | Bootstrap |
|---|---|---|---|---|
| $\left\langle V_{\Delta_{(0,\frac{1}{2})}} V_{\Delta_{(0,\frac{1}{2})}} V_{\Delta_{(0,\frac{1}{2})}} V_{\Delta_{(0,\frac{1}{2})}} \right\rangle$ | Potts | — | 4 | 4 |
| $\left\langle V_{(\frac{3}{2},0)} V_{(\frac{3}{2},\frac{2}{3})} V_{\Delta_{(0,\frac{1}{2})}} V_{\Delta_{(0,\frac{1}{2})}} \right\rangle$ | — | — | — | 5 |
| $\left\langle V_{(1,0)} V_{(1,0)} V_{(1,0)} V_{(1,0)} \right\rangle$ | $O(N)$ | 6 | — | 6 |
| $\left\langle V_{(3,0)} V_{(2,0)} V_{\Delta_{(0,\frac{1}{2})}} V_{\Delta_{(0,\frac{1}{2})}} \right\rangle$ | Potts | — | 5 | 9 |
| $\left\langle V_{(2,\frac{1}{2})} V_{(2,0)} V_{(\frac{3}{2},0)} V_{(\frac{1}{2},0)} \right\rangle$ | $O(N)$ | 77 | — | 10 |
| $\left\langle V_{(3,0)} V_{(2,\frac{1}{2})} V_{(\frac{3}{2},0)} V_{(\frac{1}{2},0)} \right\rangle$ | $O(N)$ | 660 | — | 15 |
| $\left\langle V_{(2,0)} V_{(2,0)} V_{(2,0)} V_{(2,0)} \right\rangle$ | $O(N)$, Potts | 2862 | 16 | 21 |

(3)

In the simplest cases, it is possible to identify which solutions of crossing symmetry correspond to which invariant four-tensors [3, 4]. But this does not explain the four-point functions that belong to neither the Potts nor the $O(N)$ model, because they mix fields from both. Moreover, as the fields' conformal dimensions increase, the numbers of invariant four-tensors grow much faster than the number of solutions of crossing symmetry. And it is even possible to add diagonal fields (i.e. fields with zero conformal spin) with arbitrary conformal dimensions, which belong to neither model, but lead to more solutions of crossing symmetry [5].

We will now forget about the correlation functions of one model or the other, and focus on the solutions of crossing symmetry. Of course, we will lose some information, since many different $O(N)$ four-tensors can correspond to the same solution of crossing symmetry. Shedding group-theoretic superstructures will simplify the problem, and hopefully we will be focussing on more fundamental objects. But we will have to resort to a completely different approach.

## 1.3 From loops to combinatorial maps

The $O(N)$ model and and the $Q$-state Potts models were originally defined as statistical models on lattices, with an integer parameter $N$ or $Q$. In two dimensions, they also have a loop formulation where $N$ or $Q$ takes arbitrary complex values. In this formulation, correlation functions are sums over configurations of non-intersecting loops. For example, let us draw a loop configuration on a Riemann surface of genus 2:

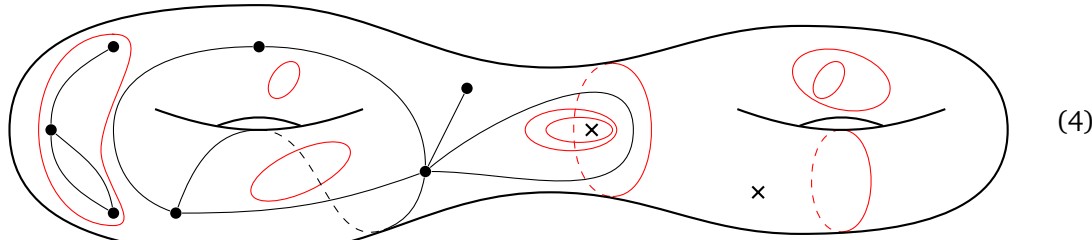

$$(4)$$

The configuration includes closed loops (in red), and segments that end at vertices. We draw a vertex as a dot if it is the endpoint of at least one segment, and as a cross otherwise. For a given correlation function, the number and valencies of vertices are fixed. On the other hand, the number of closed loops can vary a lot, depending on the configuration.

The main idea of this article is to build a correlation function by summing not over all possible loop configurations, but over a subset, defined by some constraints. These constraints must be such that the correlation function has a critical limit, is conformally invariant, and is a single-valued function of the Riemann surface's moduli. In order to satisfy these requirements, the constraints can only specify which vertices are connected by how many segments, and how these segments are ordered around each vertex. In other words, the constraints must be combinatorial.

The mathematical object that describes such constraints is a combinatorial map: basically, a graph embedded in a two-dimensional oriented manifold. Combinatorial maps appear in many contexts, and are known under various other names, such as: ribbon graphs, fatgraphs, or rotation systems. We will always use the term combinatorial map, except when making contact with a work by Do and Norbury on the moduli space of curves [6]. Since their fatgraphs are not quite the same as our combinatorial maps, but are related by graph duality, it will be convenient to keep calling them fatgraphs.

To each combinatorial map, we will associate a correlation function: a function of the vertices' positions, which also depends on the Riemann surface's moduli, and on a few other parameters. Unlike a conformal block, this is a well-defined, single-valued function. In the critical limit, it is a solution of conformal bootstrap equations.

While this function is defined unambiguously, we may be abusing terminology by calling it a correlation function, since we are not defining it as a correlation function of local fields. In fact our definition is manifestly non-local, since it relies on a combinatorial map. However, it can happen that a non-locally defined quantity can nevertheless be interpreted as a correlation function of local fields. This is the case for certain cluster connectivities of the Potts model [7], and also for some two-point functions of the loop models that we are considering [12]. It remains to be seen whether this is the case for the more general correlation functions that we are constructing.

## 1.4 Highlights of the article

- The definition (2.3) of weakly-connected combinatorial maps, which leads to simple results when it comes to counting maps, and will have a simple interpretation in conformal field theory.

- The definition (2.6) of the signature of a planar map with four vertices, which is more mysterious from the point of view of combinatorics, but which also has a simple interpretation in conformal field theory.

- The weight (44) of a loop configuration, together with a definition (45) of the angles of

edges at vertices. In contrast to the Coulomb gas formalism, our approach features an unambiguous definition of angles.

- The conjectures of Section 4.2 about the numbers of solutions of crossing symmetry equations, culminating with Conjecture 4.5 that each combinatorial map corresponds to one solution.

- Our evidence for these conjectures is summarized in Section 4.3. It consists in numerical studies of crossing symmetry in four-point functions on the sphere. We collect results from previous work on special cases, such as the $O(N)$ and Potts models, and perform a systematic scan of four-point functions in loop models.

# 2 Two-dimensional combinatorial maps

In this section, we will introduce the class of combinatorial maps that will turn out to be relevant to loop models. We will then focus on the subclass of planar maps, and review results on counting such maps. Then we will further focus on planar maps with four vertices, which will be relevant for interpreting our numerical conformal bootstrap results.

## 2.1 Connected and disconnected maps

A graph is a combinatorial object made of finitely many vertices and edges. A combinatorial map is a graph where each vertex comes with a cyclic permutation of the incident half-edges.

### 2.1.1 Connected maps

**Definition 2.1** (Connected map)
*A connected map is a connected graph, together with a cyclic permutation of the half-edges around each vertex. The corresponding compact orientable surface is built by gluing topological discs along the graph's edges, with their arrangement at each vertex determined by the cyclic permutation. These discs are the map's faces. We require that there are no monogons, i.e. that each face has at least 2 incident edge sides.*

Our definition of a combinatorial map is nonstandard in two ways: we forbid monogons, and we call a connected map what is usually called simply a map. The surface's genus $g = g_{\text{Euler}}$ can be found by computing the Euler characteristic,

$$2 - 2g_{\text{Euler}} = \#\text{vertices} - \#\text{edges} + \#\text{faces}. \tag{5}$$

If $g = 0$, the map is called planar, although it actually lives on the sphere.

We consider maps whose vertices are labelled. In practice this means that we do not identify two maps when they are related by a permutation of vertices. Let us illustrate this in an example that involves a vertex of valency 5, i.e. a vertex with 5 incident half-edges (although it only has 4 incident edges):

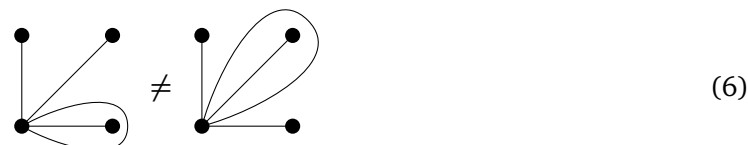

$$\tag{6}$$

We do not explicitly write labels on vertices: rather, we distinguish them by their positions in the plane, which may be considered fixed. As a result, two topologically different embeddings may be identical as maps:

$$\tag{7}$$

In this example, the first two embeddings are topologically different because they are not related by deforming their edges on the sphere. To relate them, it is necessary to move the vertices along a path with nontrivial monodromy.

Moreover, we only consider maps that do not contain monogons; equivalently we forbid edges that can be pulled inside a vertex. For example, the following two maps are the same on the sphere, and they are forbidden:

$$\tag{8}$$

On the other hand, the following map is allowed: its only face has 2 incident edge sides and is therefore not a monogon, although both edge sides belong to the same edge:

$$\tag{9}$$

### 2.1.2 Disconnected maps

Let us now consider graphs that are not necessarily connected. The surface that corresponds to a disconnected map is obtained by gluing faces from different connected components. In the case of planar maps, this just means drawing several graphs side by side, or inside one another. This allows us to have vertices of valency zero, which we write as crosses. Examples include:

$$\tag{10}$$

We may also glue a non-trivial Riemann surface to a map. This allows us to have topologically non-trivial faces. In this case, the surface's genus $g$ is not given by the Euler characteristic (5): the genera of faces have to be added,

$$g = g_{\text{Euler}} + \sum_{\text{face}} g_{\text{face}}. \tag{11}$$

For example, here is a $g = 1$ map whose tetragon face has genus one:

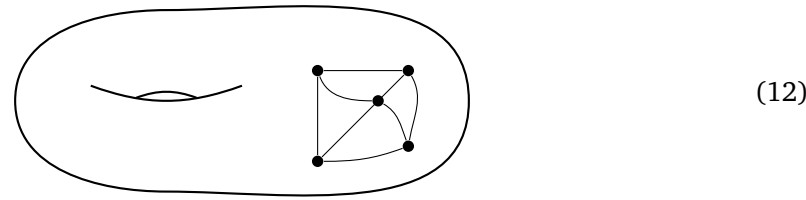

$$\tag{12}$$

In this context, our prohibition of monogons still means forbidding edges that can be pulled inside a vertex. A face is therefore not considered a monogon if it contains a handle of the Riemann surface, a connected component of the graph, or even a vertex of valency zero.

**Definition 2.2** (Set of all maps)
*For $g, n \in \mathbb{N}$, and for $r_1, r_2, \ldots, r_n \in \frac{1}{2}\mathbb{N}$ with $\sum_{i=1}^{n} r_i \in \mathbb{N}$, let $\mathcal{M}_{g,n}(r_1, r_2, \ldots, r_n)$ be the set of genus $g$ combinatorial maps without monogons, with $n$ vertices of valencies $2r_1, 2r_2, \ldots, 2r_n$, and with faces that are allowed to be topologically non-trivial.*

We write $2r \in \mathbb{N}$ for the valency of a vertex because $r$ will correspond to a Kac index of conformal field theory in Section 4. Moreover, this makes some combinatorial formulas simpler, starting with the number of edges, which is given by the handshaking lemma:

$$\boxed{\#\text{edges} = \sum_{i=1}^{n} r_i \in \mathbb{N}.} \tag{13}$$

Just like a connected map, a disconnected map is called planar if $g = 0$, i.e. if the underlying Riemann surface is the sphere. When drawing planar maps, we only draw edges and vertices, and not the sphere itself.

### 2.1.3 Weakly connected maps

We will now propose a weaker definition of connected maps, which is better adapted to map counting. It will also have a natural interpretation in conformal field theory, when it comes to decomposing correlation functions into conformal blocks, see Conjecture 4.2.

**Definition 2.3** (Set of weakly connected maps)
*Let a trivial map be one of two maps: the empty map on the sphere, and one vertex of valency zero on the sphere. Let us call splitting a map the operation of cutting the surface along a closed loop that does not intersect any edge, and eliminating the two resulting boundaries by shrinking them to nothing.*

*A map is weakly connected if it cannot be split into two non-trivial maps. For $g, n \in \mathbb{N}$, and for $r_1, r_2, \ldots, r_n \in \frac{1}{2}\mathbb{N}$ with $\sum_{i=1}^{n} r_i \in \mathbb{N}$, let $\mathcal{M}_{g,n}^c(r_1, r_2, \ldots, r_n)$ be the set of weakly connected combinatorial maps of genus $g$ without monogons, with $n$ vertices of valencies $2r_1, 2r_2, \ldots, 2r_n$.*

For example, the following map is not weakly connected, because we can split it into two non-trivial maps using the red loop:

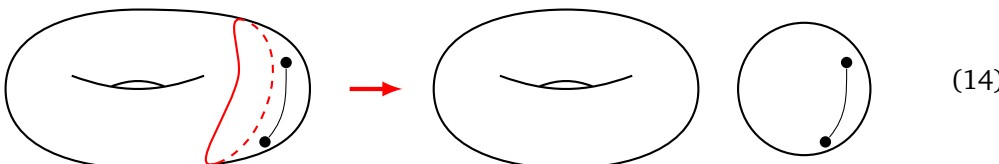

$$\tag{14}$$

This map is also not connected, because its only face contains a handle, and is therefore not topologically a disc. As the name suggests, any connected map is also weakly connected, but the reverse is not true. For example, the following planar map is weakly connected, although the underlying graph has 3 connected components:

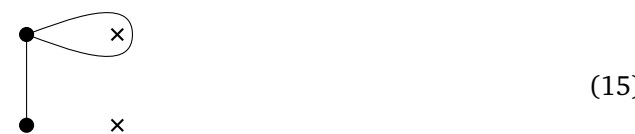

$$\tag{15}$$

## 2.2 Counting planar maps

For any values of the parameters $g, n, r_i$, we would like to determine the numbers of maps $\left|\mathcal{M}_{g,n}(r_i)\right|$ and of weakly connected maps $\left|\mathcal{M}_{g,n}^c(r_i)\right|$. We will do this in the planar case $g = 0$ by finding bijections with combinatorial sets whose numbers of elements are found in the mathematical literature.

### 2.2.1 Counting planar weakly connected maps

If a face is incident to at least two vertices of valency zero, and we split that face along a closed loop around these two vertices, we obtain two nontrivial maps, except if the original map belonged to $\mathcal{M}_{0,2}(0,0)$ or $\mathcal{M}_{0,3}(0,0,0)$. Therefore, in any weakly connected map except the sphere with two or three vertices of valency zero, a face can be incident to at most one vertex of valency zero. It follows that a planar weakly connected map can be seen as a connected map where some faces are marked, in the sense that they are incident to a vertex of valency zero. By graph duality, this is equivalent to a connected map where some vertices are marked. Since we forbid monogons, graph duality is actually a bijection between our planar weakly connected maps and the planar tight maps of [8].

Therefore, the number of weakly connected maps coincides with the number of planar tight maps $N_{0,n}(d_i)$ from [8],

$$\left|\mathcal{M}_{0,n}^c(r_1, r_2, \ldots, r_n)\right| = N_{0,n}(2r_1, 2r_2, \ldots, 2r_n). \tag{16}$$

The number of connected maps without monogons is included as the case where all $r_i$ are nonzero, since a planar connected map is nothing but a planar weakly connected map without vertices of valency zero.

Let us review some known results on these numbers: these are quasi-polynomials in $r_i^2$ of degree $n - 3$, i.e. polynomials that also depend on $r_i \bmod \mathbb{Z}$. In the case $n = 4$, the quasi-polynomial has degree one, and reads

$$\left|\mathcal{M}_{0,4}^c\right| = \left\lfloor \sum_{i=1}^4 r_i^2 - \frac{1}{2} \right\rfloor, \tag{17}$$

where we use the floor function $\lfloor x \rfloor = \max(\mathbb{N} \cap (-\infty, x])$. This formula includes three cases, depending on the number $0, 2, 4$ of indices $r_i$ in $\mathbb{N} + \frac{1}{2}$. If this number is 2 it reduces to $\sum_{i=1}^4 r_i^2 - \frac{1}{2}$, if it is 0 or 4 it reduces to $\sum_{i=1}^4 r_i^2 - 1$. For example, $\left|\mathcal{M}_{0,4}^c(2, \frac{3}{2}, 1, \frac{1}{2})\right| = 7$, and the corresponding maps are given in Eq. (A.26). Let us also give an example with vertices of valency zero: $\left|\mathcal{M}_{0,4}^c(\frac{3}{2}, \frac{3}{2}, 0, 0)\right| = 4$, and the corresponding maps are:



$$\tag{18}$$

In the case $n = 5$, the quasi-polynomial has degree two. We write three different expressions,

depending on whether $0, 2$ or $4$ arguments are half-integer:

$$\left|\mathcal{M}_{0,5}^c\right|_{r_i \in \mathbb{N}} = \frac{1}{2}\sum_{i=1}^{5} r_i^4 + 2\sum_{i<j} r_i^2 r_j^2 - \frac{5}{2}\sum_{i=1}^{5} r_i^2 + 2, \tag{19a}$$

$$\left|\mathcal{M}_{0,5}^c\right|_{r_1,r_2 \in \mathbb{N}+\frac{1}{2}} = \frac{1}{2}\sum_{i=1}^{5} \lfloor r_i^2 \rfloor^2 + 2\sum_{i<j} \lfloor r_i^2 \rfloor \lfloor r_j^2 \rfloor - \sum_{i=1}^{2} \lfloor r_i^2 \rfloor - \frac{1}{2}\sum_{i=3}^{5} r_i^2, \tag{19b}$$

$$\left|\mathcal{M}_{0,5}^c\right|_{r_1,r_2,r_3,r_4 \in \mathbb{N}+\frac{1}{2}} = \frac{1}{2}\sum_{i=1}^{5} \lfloor r_i^2 \rfloor^2 + 2\sum_{i<j} \lfloor r_i^2 \rfloor \lfloor r_j^2 \rfloor - \frac{1}{2} r_5^2. \tag{19c}$$

In these formulas, $\lfloor r^2 \rfloor = r^2 - \frac{1}{4} \in 2\mathbb{N}$ if $r \in \mathbb{N} + \frac{1}{2}$. Extracting these expressions from the general results of [8] is straightforward in principle, but tedious in practice.

### 2.2.2 Counting all planar maps

We will relate the problem of counting all planar maps to the problem of counting lattice points on the moduli space of curves, which was solved by Do and Norbury [6]. Their solution involves summing over fatgraphs, which are graph duals of our maps. More specifically, our planar maps correspond to pointed stable fatgraphs of genus zero.

Let us recall the definition of such fatgraphs from [6](Definition 2.7). Stable fatgraphs come with a genus function, which is however trivial if the genus is zero. We also simplify and correct the definition of [6] so that it corresponds to the set of graphs that is actually summed over.

**Definition 2.4** (Fatgraphs)
*A pointed stable fatgraph of genus zero is a planar fatgraph whose faces are marked, together with an equivalence relation over vertices, and a set of labels distributed on equivalence classes, such that:*

- *Any vertex of valency one is labelled or part of a non-trivial equivalence class (or both).*

- *The graph whose vertices are the connected components of our planar graph, and whose edges are defined by the equivalence relation, is a tree, i.e. a connected graph with no cycles.*

*For $n \in \mathbb{N}$, and $r_1, r_2, \ldots, r_p \in \frac{1}{2}\mathbb{N}$ with $0 \leq p \leq n$, let $\mathcal{F}_{0,n}(r_1, \ldots, r_p, 0, \ldots 0)$ be the set of pointed stable fatgraphs of genus zero with $p$ marked faces incident to $2r_1, \ldots, 2r_p$ edges, and $n-p$ labels.*

Let us illustrate this definition by enumerating all the fatgraphs in a few examples. We draw fatgraph vertices as squares, to distinguish them from the vertices of our maps:

- In the case $\left|\mathcal{F}_{0,4}(2,0,0,0)\right| = 6$, we are considering fatgraphs with one tetragonal face, and there is only one such fatgraph:

$$\tag{20}$$

Notice that the tetragonal face is twice incident to the middle vertex, and also twice incident to each one of the two edges. Since there is only one connected component, the equivalence relation must be trivial, i.e. each vertex is its own equivalence class. It remains to distribute three labels on the fatgraph, while ensuring that the two vertices of valency one have at least one label each. There are 6 ways to do so: 3 with one label on each vertex, and 3 with no label on the middle vertex. (The two outer vertices are indistinguishable.)

- In the case $\left|\mathcal{F}_{0,5}(\frac{1}{2}, \frac{1}{2}, \frac{1}{2}, \frac{1}{2}, 0)\right| = 3$, our faces are four monogons. We have to group them pairwise, and there are 3 ways to do it. This yields two connected subgraphs, with one vertex each. The two vertices must be equivalent, which we denote by a dashed line:

$$\tag{21}$$

Since there is only one equivalence class, it must be labelled.

- In the case $\left|\mathcal{F}_{0,5}(1, 1, 0, 0, 0)\right| = 16$, our faces are two digons. We can either have two connected components with one face each, or one connected fatgraph that includes both faces:

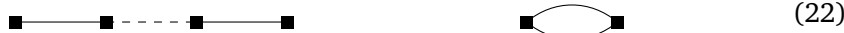

$$\tag{22}$$

On the disconnected fatgraph, we have to distribute three labels over three equivalence classes, with the two outer vertices receiving at least one label each. The two outer vertices are distinguishable, because they are associated to two different faces of the fatgraph. This leads to 12 ways of distributing the labels. (The non-trivial equivalence class receives a label in 6 cases.) On the connected fatgraph, we have to distribute three labels over two indistiguishable vertices. There are 4 ways to do it: either a vertex receives all the labels, or the labels are split over both vertices.

**Proposition 2.5** (Bijection between planar maps and planar fatgraphs)
*To any pointed stable fatgraphs of genus zero, we associate a planar map by performing graph duality on each connected component, gluing faces that are equivalent, and replacing each label with a vertex of valency zero. This is a bijection from $\mathcal{F}_{0,n}(r_1, r_2, \ldots, r_n)$ to $\mathcal{M}_{0,n}(r_1, r_2, \ldots, r_n)$.*

In particular, the absence of monogons in our maps follows from our condition on fatgraph vertices of valency one. Let us see how this bijection acts on two fatgraphs from the set $\mathcal{F}_{0,5}(1, 1, 0, 0, 0)$. We number the two faces as $1, 2$ and the three labels as $3, 4, 5$. Faces and labels correspond to the vertices of the resulting planar maps:

$$\tag{23}$$

$$\tag{24}$$

Armed with this bijection, we can use known results for sums over fatgraphs [6](Corollary 3.6). In such sums, the summand is the inverse of the order of the automorphism group of the fatgraph: however, for $n \geq 3$, automorphisms are trivial, so the sums really count fatgraphs. There is one more subtlety: some fatgraphs come with an implicit integer coefficient, which may be interpreted as an Euler characteristic. For $n = 4, 5$, this happens if the number $\#\{i | r_i = 0\}$ of vanishing parameters is large enough. If we insist on just counting fatgraphs, we have to correct the known results [6] by subtracting a function $Z_{0,n}$ of that number, which we now define:

$$Z_{0,4}(k) \underset{k \leq 3}{=} 0, \quad Z_{0,4}(4) = 1, \tag{25}$$

$$Z_{0,5}(k) \underset{k \leq 2}{=} 0, \quad Z_{0,5}(3) = 1, \quad Z_{0,5}(4) = 4, \quad Z_{0,5}(5) = 6 \tag{26}$$

(We do not have an expression for higher values of $k$ and $n$.). In particular, in the case $n = 4$, we obtain [6](Appendix A):

$$\left|\mathcal{M}_{0,4}\right|_{r_i \in \mathbb{N}} = \sum_{i=1}^{4} r_i^2 + 2 - Z_{0,4}\left(\#\{i | r_i = 0\}\right) = \sum_{i=1}^{4} r_i^2 + 2 - \prod_{i=1}^{4} \delta_{r_i,0}, \tag{27a}$$

$$\left|\mathcal{M}_{0,4}\right|_{r_1,r_2 \in \mathbb{N}+\frac{1}{2}} = \sum_{i=1}^{4} \lfloor r_i^2 \rfloor + 1, \tag{27b}$$

$$\left|\mathcal{M}_{0,4}\right|_{r_1,r_2,r_3,r_4 \in \mathbb{N}+\frac{1}{2}} = \sum_{i=1}^{4} \lfloor r_i^2 \rfloor + 3. \tag{27c}$$

Of course, this can also be obtained from the number of weakly connected maps $\left|\mathcal{M}_{0,4}^c\right|$ (17) by adding the number of maps that are not weakly connected: 3 if all $r_i$ are integer, 3 if they are all half-integer, and 1 if we have two integers and two half-integers. For $n \geq 5$, we find:

$$\left|\mathcal{M}_{0,5}\right|_{r_i \in \mathbb{N}} = \frac{1}{2}\sum_{i=1}^{5} r_i^4 + 2\sum_{i<j} r_i^2 r_j^2 + \frac{7}{2}\sum_{i=1}^{5} r_i^2 + 7 - Z_{0,5}\left(\#\{i | r_i = 0\}\right), \tag{28a}$$

$$\left|\mathcal{M}_{0,5}\right|_{r_1,r_2 \in \mathbb{N}+\frac{1}{2}} = \frac{1}{2}\sum_{i=1}^{5} \lfloor r_i^2 \rfloor^2 + 2\sum_{i<j} \lfloor r_i^2 \rfloor \lfloor r_j^2 \rfloor + 2\sum_{i=1}^{2} \lfloor r_i^2 \rfloor + \frac{3}{2}\sum_{i=3}^{5} r_i^2 + 2 - Z_{0,5}\left(\#\{i | r_i = 0\}\right), \tag{28b}$$

$$\left|\mathcal{M}_{0,5}\right|_{r_1,r_2,r_3,r_4 \in \mathbb{N}+\frac{1}{2}} = \frac{1}{2}\sum_{i=1}^{5} \lfloor r_i^2 \rfloor^2 + 2\sum_{i<j} \lfloor r_i^2 \rfloor \lfloor r_j^2 \rfloor + 3\sum_{i=1}^{4} \lfloor r_i^2 \rfloor + \frac{11}{2} r_5^2 + 3. \tag{28c}$$

Alternatively, it is possible to deduce these results from numbers of weakly connected maps (19), by adding the numbers of maps that are not weakly connected. For example, $\left|\mathcal{M}_{0,5}(\frac{3}{2}, \frac{1}{2}, 1, 1, 0)\right| = 22$ while $\left|\mathcal{M}_{0,5}^c(\frac{3}{2}, \frac{1}{2}, 1, 1, 0)\right| = 10$. The 10 weakly connected maps in this case are:

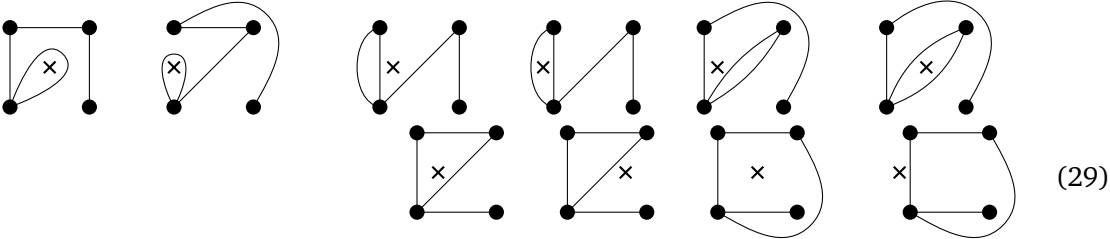

$$\tag{29}$$

The 12 maps that are not weakly connected are:

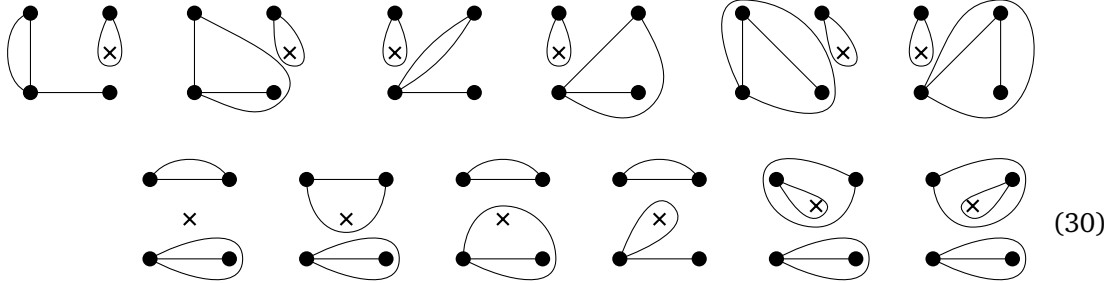

$$\tag{30}$$

For planar maps with $n \geq 6$ vertices, we expect that $\left|\mathcal{M}_{0,n}\right|$ is again given by the results in [6], minus a correction.

### 2.2.3   What about non-planar maps?

The bijection between maps and fatgraphs can be generalized to the non-planar case. In particular, maps with topologically non-trivial faces correspond to fatgraphs with a non-trivial genus function.

   The problem is however that we do not know how to count fatgraphs of nonzero genus. The weighted count of [6] is the sum of inverses of orders of fatgraphs' automorphism groups: a rational number that is in general not integer. Nontrivial automorphisms groups can only occur for $n \leq 2$: such cases are very simple if $g = 0$, but can be more complicated if $g \geq 1$.

   It may well be that the methods of [6] and/or [8] can be extended to counting non-planar maps. On the subject of counting maps, our ambition is however limited to extracting available results from the literature.

## 2.3   Planar maps with four vertices

In practice, our numerical bootstrap code only deals with four-point functions on the sphere. Therefore, planar maps with four vertices deserve special attention. We are interested not only in counting them, but also in characterizing them more finely, with the ultimate aim of associating a specific bootstrap solution to a given map.

   As a step in that direction, we will introduce the notion of the signature of a planar map with four vertices, which is inspired by the decomposition of four-point functions into conformal blocks. This notion can be generalized straightforwardly to all planar maps, and less straightforwardly to non-planar maps.

### 2.3.1   Signature of a map

**Definition 2.6** (Signature of a planar map with four vertices)
 *Given four points on the sphere, let $\mathcal{L}_s, \mathcal{L}_t, \mathcal{L}_u$ be the sets of closed loops that split the four points into $\{1,2\} \cup \{3,4\}$, $\{1,4\} \cup \{2,3\}$ and $\{1,3\} \cup \{2,4\}$ respectively. For a planar map $M$ with four numbered vertices and $x \in \{s,t,u\}$, let $e(M)$ be the union of the edges' embeddings in the sphere, and $\sigma_x(M) = \dfrac{1}{2} \min\limits_{L \in \mathcal{L}_x} |L \cap e(M)| \in \dfrac{1}{2}\mathbb{N}$. The signature of $M$ is $\sigma(M) = (\sigma_s(M), \sigma_t(M), \sigma_u(M))$.*

   For example, the following map has the signature $(1, \frac{3}{2}, \frac{3}{2})$, which we justify by drawing loops $L_s, L_t, L_u$ that minimize the numbers of intersections with the edges:

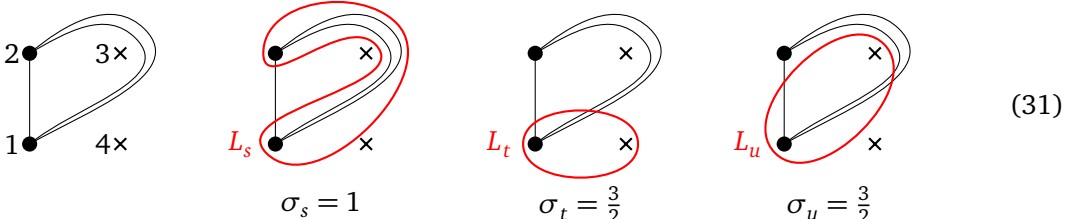

$$(31)$$

A map $M$ is weakly connected if and only if $\forall x \in \{s,t,u\}, \sigma_x(M) > 0$. To prevent a map from being weakly connected, it is enough to remove $\min_{x \in \{s,t,u\}} 2\sigma_x(M)$ edges. Sometimes, it is possible to disconnect the map by removing fewer edges: in our example, it suffices to remove the vertical edge, although $\min_{x \in \{s,t,u\}} 2\sigma_x(M) = 2$.

**Conjecture 2.7** (Total signature)
*The total signature $|\sigma(M)| = \sum_{x \in \{s,t,u\}} \sigma_x(M)$ is given by the number of edges $\sum_{i=1}^{4} r_i$, plus the number of distinct subloops $L \subset M$. A subloop is a subset of $e(M)$ that is homotopic to a circle, and has at least one vertex inside and one outside. Two subloops are considered distinct if and only if they do not intersect except possibly at vertices.*

This conjecture is motivated by inspecting examples. In the case $(r_i) = (2, 2, \frac{1}{2}, \frac{1}{2})$, let us draw three maps with 0, 1 and 2 subloops, and check that the total signature varies accordingly. For each map we indicate the total signature as $\sigma_s(M) + \sigma_t(M) + \sigma_u(M) = \sum_{i=1}^{4} r_i + \#\{\text{subloops}\}$:

$$1 + \tfrac{3}{2} + \tfrac{5}{2} = 5 + 0 \qquad 2 + \tfrac{3}{2} + \tfrac{5}{2} = 5 + 1 \qquad 3 + \tfrac{1}{2} + \tfrac{7}{2} = 5 + 2 \tag{32}$$

In the second map, we can draw two different subloops, but they are not distinct according to our definition, because they share one edge.

### 2.3.2 Counting maps with a minimum signature

**Definition 2.8** (Set of maps with a minimum signature)
*We define a partial order on signatures by $\sigma \geq \sigma' \iff \forall x \in \{s, t, u\}, \sigma_x \geq \sigma'_x$. Then for any $r_1, r_2, r_3, r_4, \sigma_s, \sigma_t, \sigma_u \in \frac{1}{2}\mathbb{N}$, the set of maps whose signature is at least $(\sigma_s, \sigma_t, \sigma_u)$ is*

$$\mathcal{M}_{0,4}(r_1, r_2, r_3, r_4 | \sigma_s, \sigma_t, \sigma_u) = \left\{ M \in \mathcal{M}_{0,4}(r_1, r_2, r_3, r_4) \middle| \sigma(M) \geq (\sigma_s, \sigma_t, \sigma_u) \right\}. \tag{33}$$

In particular, $\mathcal{M}_{0,4}\left(r_1, r_2, r_3, r_4 \middle| \frac{1}{2}, \frac{1}{2}, \frac{1}{2}\right) = \mathcal{M}_{0,4}^c(r_1, r_2, r_3, r_4)$ is the set of weakly connected maps. This definition is motivated by the conformal bootstrap approach of Section 4.2, where sets of maps with a minimum signature have a natural interpretation. This leads to a lower bound on the number of maps with a minimum signature:

**Conjecture 2.9** (Lower bound on the number of maps with a minimum signature)
*For any $r_1, r_2, r_3, r_4$ and $\sigma \geq (\frac{1}{2}, \frac{1}{2}, \frac{1}{2})$, we have*

$$\left| \mathcal{M}_{0,4}(r_1, r_2, r_3, r_4 | \sigma) \right| \geq \left| \mathcal{M}_{0,4}^c(r_1, r_2, r_3, r_4) \right| - \sum_{x \in \{s, t, u\}} \left\lfloor \left( \sigma_x - \tfrac{1}{2} \right)^2 \right\rfloor. \tag{34}$$

We do not have a combinatorial proof of this inequality, but it is obeyed in all the examples that we have considered. The term subtracted on the right-hand side involves the function

| $\sigma$ | $\frac{1}{2}$ | $1$ | $\frac{3}{2}$ | $2$ | $\frac{5}{2}$ | $3$ | $\frac{7}{2}$ | $4$ | $\frac{9}{2}$ |
|---|---|---|---|---|---|---|---|---|---|
| $\left\lfloor \left(\sigma - \frac{1}{2}\right)^2 \right\rfloor$ | 0 | 0 | 1 | 2 | 4 | 6 | 9 | 12 | 16 |

$$\tag{35}$$

Let us consider the example of $\mathcal{M}^c_{0,4}(\frac{5}{2},1,1,\frac{1}{2})$, which contains 8 maps:

$$\sigma = (\tfrac{3}{2},2,\tfrac{3}{2}) \qquad \sigma = (\tfrac{3}{2},2,\tfrac{3}{2}) \qquad \sigma = (\tfrac{5}{2},2,\tfrac{3}{2}) \qquad \sigma = (\tfrac{3}{2},2,\tfrac{5}{2})$$

$$\sigma = (\tfrac{5}{2},1,\tfrac{5}{2}) \qquad \sigma = (\tfrac{5}{2},1,\tfrac{5}{2}) \qquad \sigma = (\tfrac{1}{2},3,\tfrac{5}{2}) \qquad \sigma = (\tfrac{5}{2},3,\tfrac{1}{2})$$

(36)

The 4 maps in the rightmost two columns are uniquely characterized by their signatures, while the 4 maps in the leftmost two columns are not. The conjectured inequality is saturated for the 4 maps in the top row:

$$\left|\mathcal{M}_{0,4}\left(\tfrac{5}{2},1,1,\tfrac{1}{2}\Big|\tfrac{3}{2},2,\tfrac{3}{2}\right)\right| = 4, \tag{37}$$

$$\left|\mathcal{M}_{0,4}\left(\tfrac{5}{2},1,1,\tfrac{1}{2}\Big|\tfrac{5}{2},2,\tfrac{3}{2}\right)\right| = \left|\mathcal{M}_{0,4}\left(\tfrac{5}{2},1,1,\tfrac{1}{2}\Big|\tfrac{3}{2},2,\tfrac{5}{2}\right)\right| = 1. \tag{38}$$

On the other hand, the conjectured inequality is not saturated for the 4 maps in the bottom row:

$$\left|\mathcal{M}_{0,4}\left(\tfrac{5}{2},1,1,\tfrac{1}{2}\Big|\tfrac{5}{2},1,\tfrac{5}{2}\right)\right| = 2 > 0, \tag{39}$$

$$\left|\mathcal{M}_{0,4}\left(\tfrac{5}{2},1,1,\tfrac{1}{2}\Big|\tfrac{1}{2},3,\tfrac{5}{2}\right)\right| = \left|\mathcal{M}_{0,4}\left(\tfrac{5}{2},1,1,\tfrac{1}{2}\Big|\tfrac{5}{2},3,\tfrac{1}{2}\right)\right| = 1 > -2. \tag{40}$$

## 3 Correlation functions in loop models

In order to explain why there may be a relation between combinatorial maps and solutions of crossing symmetry, we would like to associate a correlation function to each combinatorial map. We write correlation functions as sums over loop configurations of the type

$$Z_{M,W}(\Sigma) = \sum_{E \in \mathcal{E}(\Sigma) | M(E) = M} W(E), \tag{41}$$

where:

- $M \in \mathcal{M}_{g,n}(r_1,\dots,r_n)$ is a combinatorial map of genus $g$ with $n$ vertices of valencies $2r_1,\dots 2r_n$.

- $\Sigma$ is a Riemann surface of genus $g$ with $n$ punctures.

- $\mathcal{E}(\Sigma)$ is an ensemble of configurations of non-intersecting loops on $\Sigma$, made of segments that end at punctures, together with closed loops.

- The constraint $M(E) = M$ means that the segments of the loop configuration $E$ induce the combinatorial map $M$.

- $W(E)$ is a weight function that we will define, depending on finitely many parameters.

The resulting object $Z_{M,W}(\Sigma)$ may be considered a function of the moduli of the punctured Riemann surface, parametrized by $M$ and $W$.

There are at least two methods for constructing loop ensembles $\mathcal{E}(\Sigma)$. Physicists usually work on a finite lattice, which breaks conformal symmetry but leads to finite ensembles. Mathematicians have introduced conformal loop ensembles, which preserve conformal symmetry but are infinite. We will remain agnostic on the definition of $\mathcal{E}(\Sigma)$, and focus on determining loop weights that preserve conformal symmetry. After that, we will discuss how to approximate $Z_{M,W}(\Sigma)$ on a lattice. But we will not prove that our lattice sums have well-defined continuum limits, and our construction remains conjectural.

## 3.1 Weights of loop configurations

An important building block of the weight function $W(E)$ on the ensemble of loop configurations is a weight function $w(C)$ on the set of closed loops.

### 3.1.1 Weights of closed loops

**Definition 3.1** (Combinatorial signature of a loop in a punctured Riemann surface)
*Let $\Sigma$ be a genus $g$ Riemann surface with $n$ punctures labelled $1,\dots,n$. Let $C$ be a closed loop on $\Sigma$. We define the combinatorial signature of $C$ as*

$$\chi_\Sigma(C) = \begin{cases} (-1, \emptyset), & \text{if } C \text{ cuts a handle but keeps } \Sigma \text{ connected,} \\ (g', I), & \text{with } 1 \notin I \subset \{1, \dots, n\}, \text{ if } \Sigma - C \text{ is not connected,} \end{cases} \tag{42}$$

*where in the last case $g'$ is the genus and $I$ the set of punctures of the connected component that does not contain the first puncture.*

For example, here are four loops with their combinatorial signatures:

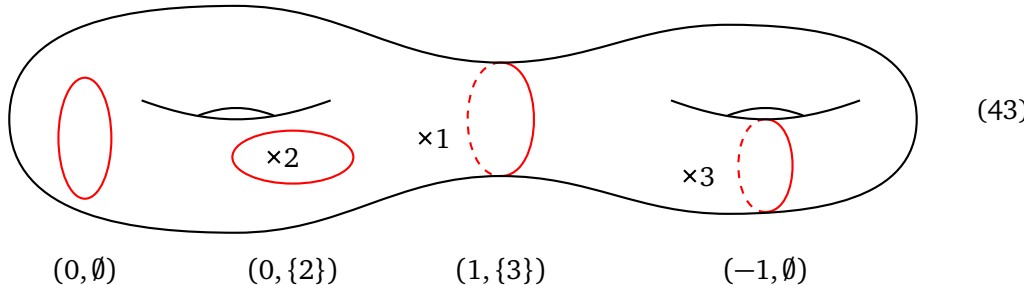

$$(0, \emptyset) \qquad (0, \{2\}) \qquad (1, \{3\}) \qquad (-1, \emptyset) \tag{43}$$

Then a loop weight is a function on the image $\operatorname{Im} \chi_\Sigma$, equivalently a function on the set of loops that only depends on their combinatorial properties. This is stronger than requiring that it only depends on the topology of loops: for example, on a torus with no punctures, there are infinitely many topologically distinct closed loops, but only two possible combinatorial signatures, namely $(-1, \emptyset)$ and $(0, \emptyset)$.

For example, in the case $g = 0$ and $r_i = 0$, $\operatorname{Im} \chi_\Sigma$ is made of the subsets of $\{2, \dots, n\}$, so that $w \in \mathbb{C}^{2^{n-1}}$. Moreover, there is only one map in $\mathcal{M}_{0,n}(0, \dots, 0)$. We therefore recover the fact that there is one $n$-point function of diagonal fields for any $w \in \mathbb{C}^{2^{n-1}}$ [5,9].

Now, in loop configurations such that $M(E) = M$, not all closed loops are allowed. To begin with, there can be no closed loop around one vertex whose valency is not zero. Moreover, the signature $\chi_\Sigma(C) = (g', I)$ obeys $\sum_{i \in I} r_i \in \mathbb{N}$, i.e. the loop $C$ preserves the conservation of $r$ modulo integers (13). These conditions depend only on the valencies $r_i$, and not on the particular choice of the combinatorial map $M$. The weight function $w : \operatorname{Im} \chi_\Sigma \to \mathbb{C}$ needs be defined only over allowed loops.

### 3.1.2 Dependence on local angles at vertices

The constraint $M(E) = M$ and the weights of closed loops depend only on the combinatorial properties of loop configurations. This is necessary for the sum $Z_{M,E}(\Sigma)$ to be conformally invariant in the critical limit, and a single-valued function of the moduli of $\Sigma$. However, conformal symmetry and single-valuedness allow loop weights to depend on local angles. The weight of a loop configuration can therefore depend on the relative angles $\theta_{i,1}, \dots, \theta_{i,2r_i}$ of the $2r_i$ segments that meet at $z_i$.

Such angles are well-defined provided the segments are differentiable at $z_i$. However, assuming differentiability would be too strong a restriction on the ensemble $\mathcal{E}(\Sigma)$, as we expect typical loop configurations to be fractal and therefore far from differentiable. Actually, we do not need an unambiguous definition of angles in individual loop configurations: only the sum over configurations needs be well-defined. We will propose how to achieve this on the lattice in Section 3.2. For the moment, we will assume that the local angles $\theta_{i,1}, \dots, \theta_{i,2r_i}$ make sense.

We then introduce an angular momentum $s_i$ (sometimes called pseudo-momentum [1]) at each vertex, and write the weight of a loop configuration $E$ as

$$
W(E) = \prod_{i=1}^{n} \exp\left\{\tfrac{i}{2} s_i \left(\sum_{k_i=1}^{2r_i} \theta_{i,k_i}\right)\right\} \prod_{C \in E} w(\chi_\Sigma(C)), \tag{44}
$$

where the second product is over the closed loops that belong to $E$. Writing this formula is pretty straightforward: the subtle issue is to choose reference directions for the angles $\theta_{i,1}, \dots, \theta_{i,2r_i}$, such that the weights $W(E)$ do not depend on this choice.

To define the angles, we will actually introduce not only a reference direction at each vertex, but also a reference edge, which we call edge number 1. We assume that its angle obeys $\theta_{i,1} \in [0, 2\pi)$, and further assume

$$
\boxed{\theta_{i,k_i} \in [\theta_{i,1}, \theta_{i,1} + 2\pi)} \tag{45}
$$

These conventions take into account the cyclic ordering of the half-edges around our vertex. Thanks to this ordering, the angles are a priori not defined under individual shifts $\theta_{i,k} \to \theta_{i,k} + 2\pi m_k$ with $(m_k) \in \mathbb{Z}^{2r_i}$, but under a global shift $\theta_{i,k} \to \theta_{i,k} + 2\pi m$ with $m \in \mathbb{Z}$. In terms of the angular momentum, this implies $s_i \in \frac{1}{r_i}\mathbb{Z}$.

Any change of the reference edge or reference direction modifies the weights $W(E)$ by an $E$-independent factor. In particular, a small change $\delta\theta_i$ of the reference direction leads to $\theta_{i,k} \to \theta_{i,k} - \delta\theta_i$ for any $k$, except if the reference direction crosses the reference edge i.e. $\theta_{i,1} - \delta\theta_i < 0$. In this case we have $\theta_{i,k} \to \theta_{i,k} - \delta\theta_i + 2\pi$. In both cases, the weights are multiplied with an $E$-independent phase $W(E) \to e^{-i r_i s_i \delta\theta_i} W(E)$. Similarly, changing the reference edge from one edge to the next leads to $W(E) \to e^{i\pi s_i} W(E)$.

Before discussing subtleties with the choice of a reference edge, let us summarize the parameters of the sum $Z_{M,W}(\Sigma)$:

- The combinatorial map $M$, including the valencies $2r_i \in \mathbb{N}$.

- The weight function $W$ depends on an angular momentum $s_i \in \frac{1}{r_i}\mathbb{Z}$ at each vertex.

- The weight function $W$ also depends on a number of continuous parameters: the weights of closed loops. There is at least 1 such parameter (the weight of contractible loops), and at most $2^{n-1}$ if $g = 0$.

### 3.1.3 Reference edges for computing weights

We need to single out a reference edge for each vertex in a given combinatorial map $M$. This is a priori not trivial, because the edges are not marked. Nevertheless, in most combinatorial maps, it is possible to mark edges, using which vertices they connect — even when all edges from a vertex connect to the same other vertex, for example:

$$\tag{46}$$

In the first example, the edges that end at vertex 3 all lead to vertex 1, but they can be distinguished from one another by their order around vertex 1, relative to the edges $1-2$ and $1-4$. In the second example, the three edges $1-2$ are distinguished from one another by the presence of two vertices of valency zero on one face.

If however the combinatorial map $M$ has a nontrivial symmetry, we cannot mark edges. The simplest example is the two-point function on the sphere:

$$\tag{47}$$

In this case, instead of marking an edge, we can assume that the same edge serves as a reference for both vertices. Changing this edge to the next one leads to a phase $e^{i\pi(s_1-s_2)}$, which is trivial if $s_1 \equiv s_2 \bmod 2$. We take this as a necessary condition for the two-point function on the sphere to be nonzero, in addition to the obvious condition $r_1 = r_2$. These combinatorial conditions are a bit weaker than the constraint $(r_1, s_1) = (r_2, s_2)$ that the two fields have the same conformal dimensions.

There are other combinatorial maps whose symmetries prevent us from marking edges, for example:

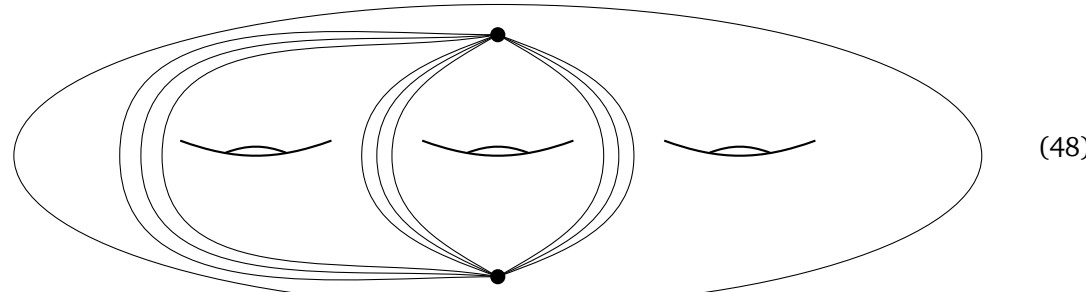

$$\tag{48}$$

The $\mathbb{Z}_3$ symmetry of this map allows us to change the reference edge to the next-to-next-to-next edge, leading to a phase $e^{3i\pi(s_1-s_2)}$. We require this phase to be trivial, which leads to the condition

$$s_1 - s_2 \in \frac{2}{3}\mathbb{Z}, \tag{49}$$

while $s_i \in \frac{2}{9}\mathbb{Z}$. If this condition is violated, our combinatorial map does not lead to a correlation function. Some other combinatorial maps in $\mathcal{M}_{3,2}(1,1)$ do not have the $\mathbb{Z}_3$ symmetry, and lead to nonzero correlation functions, with no conditions on $s_i$ beyond $s_i \in \frac{1}{r_i}\mathbb{Z}$.

## 3.2 Lattice approximation

Let us indicate how correlation functions can be computed on a lattice. In particular, we will discuss how to represent vertices of arbitrary valencies, and how to compute angles. After that, we will compare our lattice sums with previous work on loop models.

### 3.2.1 Valencies and angles on a square lattice

For simplicity, we consider a square lattice. In this lattice, each node has valency 4, so it seems difficult to draw a combinatorial map with vertices of valencies $> 4$. The well-known solution is for a vertex of the combinatorial map to be represented by several nodes of the lattice:

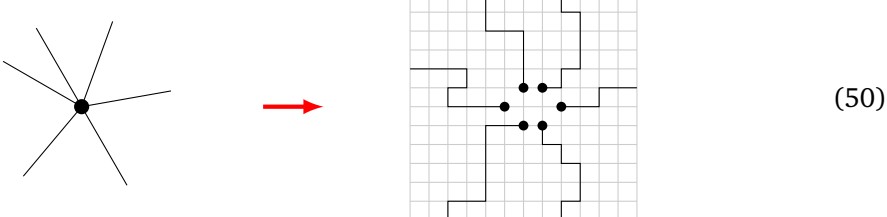

(50)

In the critical limit, the lattice spacing goes to zero. If they are kept at finitely many lattice steps from one another, our 6 nodes coincide in the critical limit, and may be considered as one and the same vertex.

Similarly, lattice angles belong to $\{0, \frac{\pi}{2}, \pi, \frac{3\pi}{2}\}$. To represent more general angles, we should not focus on one lattice node, but consider a larger region — say, compute the angle of an edge at the first point where that edge intersects the circle of radius $\ell$ lattice steps centered at the node:

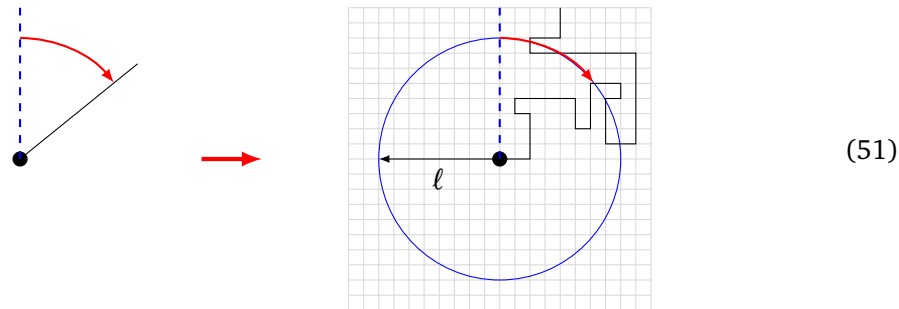

(51)

For a lattice of size $L$, with $L \to \infty$ in the critical limit, we define local angles by taking

$$1 \ll \ell \ll L. \qquad (52)$$

We conjecture that with this definition of angles, the lattice version $Z_{L,\ell}(\mathcal{E}, f, w)$ of the sum over configurations (41) has a well-defined limit, up to a simple rescaling. More specifically, there exists a function $\phi(L, g, n)$ of the lattice size $L$, genus $g$ and number of punctures $n$, such that $\lim_{L \to \infty} \phi(L, g, n) Z_{L,\ell}(\mathcal{E}, f, w) \in \mathbb{C}^*$. In particular, since $\phi(L, g, n)$ does not depend on the positions $z_i$ of the punctures, we obtain a non-trivial function of these positions, which is defined up to a $z_i$-independent factor.

Our definition of angles does not contradict the expectation that loops and segments are fractal, and therefore non-differentiable. We are not defining angles for individual segments: actually, we are not even trying to follow a given segment when we vary the lattice size. We are only conjecturing that a sum over an ensemble of loop configurations has a finite limit.

### 3.2.2 Comparison with the Coulomb gas approach

In the Coulomb gas approach to the $O(N)$ model [10], oriented loops are viewed as domain walls between regions of different heights $h$ in a solid-on-solid model. The heights of two adjacent regions differ by $\pm h_0$, where the sign determines the orientation of the corresponding loop. The value of the height field at a given point can be obtained by counting (algebraically) the number of walls encountered when going from this point to the boundary of the system. The fugacity $N$ of closed loops is reproduced by assigning a weight $e^{\frac{i}{2}\theta s_0}$ to each turn of a domain wall by an angle $\theta$. Since closed loops on the sphere turn by a total angle $\pm 2\pi$, the parameter $s_0$ should be chosen such that $N = e^{\pi i s_0} + e^{-\pi i s_0}$.

This approach can describe not only closed loops, but also lines from a vertex to another vertex, which correspond to the segments of our loop ensembles, i.e. to the edges of our combinatorial maps. Such lines are also associated with dislocations of the height field. A line that turns by an angle $\Theta$ from a vertex to another vertex again gives rise to a weight $e^{\frac{i}{2}\Theta s_0}$.

It is possible to trade angles for heights. For example, when a line winds $p$ times around a vertex, the height at the vertex increases by $\pm p h_0$ compared to the height at infinity, since we cross $p$ walls to reach the vertex along some given straight line:

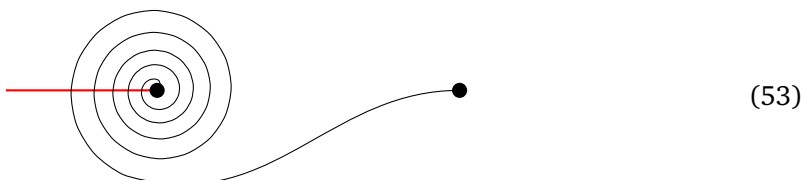
(53)

The height associated to an angle $\theta$ is in general $h_\theta = \left\lfloor \frac{\theta}{2\pi} \right\rfloor h_0$, and we are doing the replacement $e^{\frac{i}{2}\theta s_0} \to e^{i\pi \frac{h_\theta}{h_0} s_0}$, or $e^{\frac{i}{2}\Theta s_0} \to e^{i\pi \frac{h(z_1)-h(z_2)}{h_0} s_0}$ for a line that joins two vertices at positions $z_1$ and $z_2$. Some information is lost when truncating angles to integer multiples of $2\pi$, but this does not matter when doing statistics over large angles.

Heights can be ambiguous. In our example, the red straight line from the left crosses $p = 5$ walls, but from a different orientation it could be $p = 4$. Worse, it is actually possible to reach the vertex from infinity without crossing any line, by following a spiral. These ambiguities mean that the height field is not single-valued.

Nevertheless, the Coulomb gas approach, which treats the height field as a free boson and builds the $O(N)$ model as a perturbed free boson theory, gives correct results for scaling dimensions [10]. The approach is able to describe magnetic operators, which create dislocations (our vertices with indices $(r, 0)$), electric operators, which are exponentials of the height field (our vertices of valency zero), and electro-magnetic operators (vertices with nonzero indices $(r, s)$) [11]. The Coulomb gas approach can be used for deriving the winding angle distribution for self-avoiding random walks [12], and its generalisation to self-avoiding stars [13]. All these well-established results confirm the validity of the approach [14].

The Coulomb gas calculations make crucial use of the lattice approximation. In particular, in the case of the Brownian motion, computing probability distributions directly in the continuum gives rise to unphysical divergences [15], which can be cured by the lattice approximation. (See [16] for a recent review.) It is expected that this problem does not arise for self-avoiding walks (or, more generally, in the $O(N)$ loop model), because such walks cannot visit the same point several times. This suggests that a genuine definition of angles in the continuum may be possible.

In the present article, we have proposed an unambiguous definition of correlation functions in loop models, including in particular their dependence on angles, without the need for a height field. This definition can be approximated on a lattice, but we hope that it also makes

sense in the continuum. While our approach is broadly consistent with the Coulomb gas approach, it is not obvious to us that the two approaches are equivalent, and it is not clear that the Coulomb gas approach can provide a satisfactory definition of the whole set of correlation functions that we have built from combinatorial maps.

# 4 Conformal bootstrap

With our construction of correlation functions in loop models, we have shown that combinatorial maps may well parametrize correlation functions in conformal field theory. But in which conformal field theory? This is the question that we will now address, using the conformal bootstrap approach.

## 4.1 Models and correlation functions

In the bootstrap approach, a correlation function may be characterized as a solution of linear equations such as crossing symmetry or modular invariance. A model or theory is a set of correlation functions that are related by non-linear equations, which we will call factorization constraints. After reviewing these notions, we will introduce the correlation functions and the models we are interested in.

### 4.1.1 Crossing symmetry and factorization

For technical simplicity, we now focus on four-point functions on the sphere. From the axiom of the existence and convergence of operator product expansions, it follows that a four-point function $\left\langle \prod_{i=1}^{4} V_{\delta_i}(z_i) \right\rangle$ has three equivalent decompositions into conformal blocks $\mathcal{G}_{\delta}^{(x)}(c|\delta_i|z_i)$, called the $s$-channel, $t$-channel and $u$-channel decompositions. Schematically,

$$\left\langle \prod_{i=1}^{4} V_{\delta_i}(z_i) \right\rangle = \sum_{\delta \in \mathcal{S}^{(x)}} D_{\delta}^{(x)}(c|\delta_i) \, \mathcal{G}_{\delta}^{(x)}(c|\delta_i|z_i), \quad \forall \, x \in \{s, t, u\}, \tag{54}$$

where $\mathcal{S}^{(x)}$ is the $x$-channel spectrum, and $D_{\delta}^{(x)}(c|\delta_i)$ an $x$-channel four-point structure constant.

Given the spectra, we therefore have a linear system of equations for the four-point structure constants, called crossing symmetry equations. This linear system only depends on the representations of the conformal algebra that appear in our spectra and four-point function. Here we parametrize a representation of the conformal algebra by a pair $\delta = (\Delta, \bar{\Delta})$ of left- and right-moving conformal dimensions of a primary state. This primary state generates the representation when it is a highest-weight representation. Loop models also involve logarithmic representations, which are not generated by their primary states. This technical subtlety has been thoroughly dispatched in previous work [3,17], and plays no role in our analysis: we will treat logarithmic representations on the same footing as highest-weight representations.

In loop models, the linear system (54) turns out to have a finite-dimensional space of solutions, whose dimension does not depend on the central charge $c$, which can vary continuously. We write this dimension as

$$d_{0,4}(\delta_i|\mathcal{S}^{(x)}) = \dim\left\{\left(D_{\delta}^{(s)}, D_{\delta}^{(t)}, D_{\delta}^{(u)}\right)\middle|(54)\right\}. \tag{55}$$

We could similarly define the dimensions $d_{g,n}$ of spaces of solutions of the conformal bootstrap equations for $n$-point functions on a Riemann surface of genus $g$.

Operator product expansions do not just imply linear crossing symmetry equations: they also imply that four-point structure constants factorize into three-point structure constants. A major complication, which does occur in loop models, is the existence of non-trivial field multiplicities, i.e. the possibility that several different fields share the same dimensions $\delta$. Calling $m_\delta \in \mathbb{N}$ this multiplicity, the factorization constraint reads

$$D_\delta^{(s)}(\delta_1, \delta_2, \delta_3, \delta_4) = \sum_{k=1}^{m_\delta} C\big(\delta_1, \delta_2, (\delta, k)\big) C\big(\delta_3, \delta_4, (\delta, k)\big), \tag{56}$$

where for simplicity we omit the dependence on the central charge $c$, and the possible multiplicities of the fields with dimensions $\delta_1, \delta_2, \delta_3, \delta_4$. Field multiplicities can also manifest themselves by the existence of larger numbers of crossing symmetry solutions for four-point functions $\left\langle \prod_{i=1}^4 V_{\delta_i}(z_i) \right\rangle$, since the solutions that correspond to different fields $V_{\delta_1}(z_1)$ (say) with the same dimension $\delta_1$ can be linearly independent — although this is not always true [3].

**Definition 4.1** (Conformal field theory)
*A consistent conformal field theory on the plane is a set of correlation functions that obey crossing symmetry and factorization, such that for any representation $\delta$ that appears in the spectrum of a correlation function, there exist correlation functions of the corresponding field $V_\delta$.*

### 4.1.2 Loop models and their spectra

We would like to define a loop model as a set of correlation functions that includes those of the $O(N)$ model [3] and of the Potts model [4]. It should also include correlation functions that mix fields from both models, and correlation functions that involve arbitrary diagonal fields [5].

A loop model depends on a parameter $\beta^2 \in \mathbb{C}$ which is related to the central charge by

$$c = 13 - 6\beta^2 - 6\beta^{-2}, \tag{57}$$

and which obeys the constraint

$$\Re\beta^2 > 0. \tag{58}$$

In terms of $\beta^2$, the Kac table conformal dimensions are given in terms of Kac table indices $(r, s)$ by

$$\Delta_{(r,s)} = \frac{1}{4}\big(r\beta - s\beta^{-1}\big)^2 - \frac{1}{4}\big(\beta - \beta^{-1}\big)^2. \tag{59}$$

We introduce the following notations and terminology for primary fields and their left- and right-conformal dimensions $\delta = (\Delta, \bar\Delta)$:

| Name | Non-diagonal | Degenerate | Diagonal | |
|---|---|---|---|---|
| Fields | $V_{(r,s)}$ | $V_{\langle 1,s \rangle}$ | $V_\Delta$ | |
| Dimensions | $\delta_{(r,s)} = \big(\Delta_{(r,s)}, \Delta_{(r,-s)}\big)$ | $\delta_{\langle 1,s \rangle} = \big(\Delta_{(1,s)}, \Delta_{(1,s)}\big)$ | $\delta_\Delta = (\Delta, \Delta)$ | (60) |
| Parameters | $r \in \frac{1}{2}\mathbb{N}^*,\ s \in \frac{1}{r}\mathbb{Z}$ | $s \in \mathbb{N}^*$ | $\Delta \in \mathbb{C}$ | |
| Restrictions | $-1 < s \le 1$ | $s \in \{2, 3\}$ | None | |

An essential structural feature of loop models is the existence of the degenerate field $V_{\langle 1,3 \rangle}$. This leads to constraints on structure constants, which may be considered as a symmetry called interchiral symmetry [18]. In practice, interchiral symmetry determines how structure constants behave under the shift $s \to s+2$ of the second Kac table index. This allows us to impose restrictions on the values of $s$ for non-diagonal and degenerate fields, see Table (60). In the

case of diagonal fields, interchiral symmetry determines how structure constants behave under $P \to P + \beta^{-1}$, where the momentum $P$ is defined by

$$\Delta = \Delta_{(0,0)} + P^2 . \tag{61}$$

In crossing symmetry equations (54), we can then replace conformal blocks with infinite linear combinations called interchiral blocks. For example, in the case of a non-diagonal field, an interchiral block reads

$$\widetilde{\mathcal{G}}^{(x)}_{\delta_{(r,s)}} = \sum_{s' \in s + 2\mathbb{Z}} c^{(x)}_{(r,s')} \mathcal{G}^{(x)}_{\delta_{(r,s')}} , \tag{62}$$

for some coefficients $c^{(x)}_{(r,s')}$ that are explicitly known [3].

For $r_0 \in \frac{1}{2}\mathbb{N}^*$, we introduce the set of non-diagonal fields whose first Kac index is no less than $r_0$, and differs from $r_0$ by an integer:

$$\boxed{\mathcal{S}_{r_0} = \left\{ \delta_{(r,s)} \,\middle|\, r \in r_0 + \mathbb{N}, s \in \frac{1}{r}\mathbb{Z} \cap (-1,1] \right\} .} \tag{63}$$

With this notation, let us write the spectra of the $O(N)$ and Potts models, and the family of spectra that appear in four-point functions of diagonal fields [5]:

$$\mathcal{S}^{O(N)} = \mathcal{S}_{\frac{1}{2}} \cup \mathcal{S}_1 \cup \left\{ \delta_{\langle 1,3 \rangle} \right\} , \tag{64}$$

$$\mathcal{S}^{\text{Potts}} = \mathcal{S}_2 \cup \left\{ \delta_{\langle 1,2 \rangle}, \delta_{\langle 1,3 \rangle} \right\} \cup \left\{ \delta_{\Delta_{(0,\frac{1}{2})}} \right\} , \tag{65}$$

$$\mathcal{S}^{\Delta} = \mathcal{S}_1 \cup \{ \delta_{\Delta} \} . \tag{66}$$

These spectra are infinite but discrete: in particular, while $\Delta$ may take arbitrary complex values, a particular spectrum $\mathcal{S}^{\Delta}$ only involves one value.

### 4.1.3 Correlation functions

We will consider correlation functions of non-diagonal and/or diagonal fields,

$$\left\langle \prod_{i=1}^d V_{\Delta_i}(z_i) \prod_{i=d+1}^n V_{(r_i,s_i)}(z_i) \right\rangle . \tag{67}$$

We do not include degenerate fields, whose correlation functions are severely constrained by BPZ differential equations. Nevertheless, the existence of degenerate fields of the type $V_{\langle 1,s \rangle}$ implies the conservation of the first Kac index $r$ modulo integers [19], where by convention a diagonal field $V_{\Delta}(z)$ has $r = 0$. Given the identification of $r$ with half the number of edges in a combinatorial map, this constraint is formally identical to the requirement (13) that the number of edges be integer.

Let us focus on a four-point function, and its spectra $\mathcal{S}^{(s)}, \mathcal{S}^{(t)}, \mathcal{S}^{(u)}$. The constraint (13) also applies to these spectra, for examples the fields in $\mathcal{S}^{(s)}$ must have $r \in r_1 + r_2 + \mathbb{Z}$. In practice, if we include fields that violate this constraint, we find that their structure constants vanish when we compute them by solving crossing symmetry. Similarly, the presence of degenerate fields is constrained by the fusion rules

$$V_{\langle 1,s \rangle} \in V_{(r_1,s_1)} \times V_{(r_2,s_2)} \implies \begin{cases} r_1 = r_2 , \\ s \in |s_1 - s_2| + 1 + 2\mathbb{N} . \end{cases} \tag{68}$$

Taking interchiral symmetry into account, this constraint reduces to $r_1 = r_2$ and $s \equiv s_1 - s_2 + 1 \bmod 2$.

Let us consider the case where we build the $s$-channel, $t$-channel and $u$-channel spectra by allowing all non-diagonal fields that respect constraint (13). Then we denote the dimension of the space of solutions of crossing symmetry as

$$d_{0,4}^c(\delta_i) = d_{0,4}\left(\delta_i\middle|\mathcal{S}_{r_1+r_2 \bmod 1}, \mathcal{S}_{r_1+r_4 \bmod 1}, \mathcal{S}_{r_1+r_3 \bmod 1}\right), \tag{69}$$

where by convention $r \bmod 1 \in \{\frac{1}{2}, 1\}$. We also consider the case where we add one diagonal field in each channel where the constraint (13) allows it. This amounts to adding 1 or 3 diagonal fields, depending on $r_i \bmod 1$. Calling $d_{0,4}(\delta_i)$ the dimension of the space of solutions, we have for example

$$d_{0,4}(\delta_i) \underset{r_1 \equiv r_2 \equiv r_3 \equiv r_4 \bmod 1}{=} d_{0,4}\left(\delta_i\middle|\mathcal{S}^{\Delta_s}, \mathcal{S}^{\Delta_t}, \mathcal{S}^{\Delta_u}\right), \tag{70}$$

$$d_{0,4}(\delta_i) \underset{r_1+\frac{1}{2} \equiv r_2+\frac{1}{2} \equiv r_3 \equiv r_4 \bmod 1}{=} d_{0,4}\left(\delta_i\middle|\mathcal{S}^{\Delta_s}, \mathcal{S}_{\frac{1}{2}}, \mathcal{S}_{\frac{1}{2}}\right), \tag{71}$$

for any $\Delta_s, \Delta_t, \Delta_u \in \mathbb{C}$.

## 4.2 Solutions of conformal bootstrap equations

### 4.2.1 Main conjectures

**Conjecture 4.2** (Correlation functions without diagonal fields in the spectrum)
*For any $n$-point function of diagonal and non-diagonal fields on a Riemann surface of genus $g$, the dimension of the space of solutions of conformal bootstrap equations with spectra made only of non-diagonal fields is the number of weakly connected maps $\left|\mathcal{M}_{g,n}^c(r_i)\right|$. In particular,*

$$d_{0,4}^c(\delta_i) = \left|\mathcal{M}_{0,4}^c(r_i)\right|. \tag{72}$$

This conjecture justifies our definition 2.3 of weakly connected maps. In contrast to connected maps, weakly connected maps can have vertices of valency zero, so they allow us to account for correlation functions that involve diagonal fields.

**Conjecture 4.3** (Correlation functions with diagonal fields in the spectrum)
*For any $n$-point function of diagonal and non-diagonal fields on a Riemann surface of genus $g$, the dimension of the space of solutions of conformal bootstrap equations with spectra made of non-diagonal fields, plus one diagonal or degenerate field whenever allowed by fusion rules, is the number of maps $\left|\mathcal{M}_{g,n}(r_i)\right|$. In particular,*

$$d_{0,4}(\delta_i) = \left|\mathcal{M}_{0,4}(r_i)\right|. \tag{73}$$

As they are written, these conjectures are only supposed to be true if $\mathcal{M}_{g,n}(r_i)$ does not include maps with non-trivial symmetries. As we discussed at the end of Section 3.1, non-trivial symmetries can indeed lead to some would-be correlation functions actually vanishing, depending on the values of the second indices $s_i$ of the non-diagonal fields. Maps with symmetries are rare, in particular they occur only if $n \leq 2$. If $\mathcal{M}_{g,n}(r_i)$ includes maps with symmetries, the conjectures are still valid if $s_i$ obeys conditions of the type (49), including in particular if $s_i = 0$. If however these conditions are violated, we have to remove the corresponding maps from $\mathcal{M}_{g,n}(r_i)$, and we predict fewer bootstrap solutions.

### 4.2.2 Further conjectures for four-point functions on the sphere

Our conjectures are only valid for spectra that include all non-diagonal fields that are allowed by the $r$-conservation condition (13), plus possibly one diagonal or degenerate field. It is also interesting to count correlation functions with more general spectra.

The general idea is that removing one field from a spectrum $\mathcal{S}^{(x)}$ in one channel removes one correlation function, since we are imposing a linear constraint $D_{\delta}^{(x)} = 0$ in our system of crossing symmetry equations. However, it can happen that $D_{\delta}^{(x)} = 0$ already holds for all solutions, in which case imposing it as a constraint changes nothing [3]. Conversely, adding one field generally adds one correlation function, provided the field is of the type $V_{(r,s)}, V_{\langle 1,s\rangle}$ or $V_\Delta$, respects fusion rules including $r$-conservation, and was absent from the original spectrum. Again, exceptions can occur. However, we conjecture that whenever we add/remove one diagonal field $V_\Delta$ with a generic conformal dimension (say $\Delta \neq \Delta_{(r,s)}$ for any $r, s \in \mathbb{Q}$), we gain/lose one correlation function.

Consider the identification of the first Kac index $r$ of a non-diagonal field, with half the number of edges at the corresponding vertex. So far, we have been applying this identification to the fields in our correlation function (67): what about fields in the spectra $\mathcal{S}^{(x)}$? The idea is now that the number of edges in the channel $x$, as measured by the signature $\sigma_x$ of Definition 2.6, corresponds to the smallest $r$-index in $\mathcal{S}^{(x)}$.

**Conjecture 4.4** (Four-point functions with smaller spectra)
*For any four-point of diagonal and non-diagonal fields on the sphere, the dimension of the space of solutions of crossing symmetry with spectra of the type $\mathcal{S}_{r_0}$ (63) is the number of combinatorial maps with a minimum signature,*

$$d_{0,4}\left(\delta_i \big| \mathcal{S}_{\sigma_s}, \mathcal{S}_{\sigma_t}, \mathcal{S}_{\sigma_u}\right) = \left|\mathcal{M}_{0,4}(r_i | \sigma_s, \sigma_t, \sigma_u)\right| . \tag{74}$$

*In the case $\sigma_x = 0$, we define $\mathcal{S}_0 = \mathcal{S}^\Delta$ (66) for some arbitrary dimension $\Delta$.*

This reduces to Conjecture 4.2 if $\sigma_x \in \{\frac{1}{2}, 1\}$, and to Conjecture 4.3 if $\sigma_x \in \{0, \frac{1}{2}\}$. Quite curiously, this conjecture has purely combinatorial consequences when it comes to counting maps with a minimum signature. Reducing the spectrum from $\mathcal{S}_{\frac{1}{2}}$ or $\mathcal{S}_1$ to $\mathcal{S}_\sigma$ indeed amounts to setting $\left\lfloor (\sigma - \frac{1}{2})^2 \right\rfloor$ (35) structure constants to zero. For each one of these extra constraints, we lose at most one solution. This leads to an inequality on two numbers of solutions that are both given by combinatorial formulas according to Conjectures 4.2 and 4.4. As a corollary, we obtain Conjecture 2.9 for numbers of maps with a minimum signature.

### 4.2.3 Relation with loop models

Let us first indicate how the parameters of loop models are related to the parameters of conformal field theory. We now summarize the relations, before discussing them in more detail:

| Loop models | Conformal field theory | |
|---|---|---|
| $2r = $ vertex valency<br>$s = $ vertex angular momentum | $(r,s) = $ Kac table indices of $V_{(r,s)}$ | (75) |
| $w$(contractible loop) | $-2\cos(\pi\beta^2) \rightarrow$ central charge | |
| $w$(non-contractible loop) | $2\cos(2\pi\beta P) \rightarrow$ dimension of $V_\Delta$ | |

To begin with, each vertex comes with a valency $2r$ and an angular momentum $s$, which are identified with the Kac table indices $(r, s)$ of conformal field theory. This identification has

been consistently observed in various approaches to loop models, including most recently in an integrable approach [20].

Then let us consider the weight function $w(C)$, which is defined on closed loops. For contractible loops, the value of this function is related to the central charge (57). In the $O(N)$ model and $Q$-state Potts model, the weight of contractible loops is given in terms of the model's parameter by $w = N$ and $w = \sqrt{Q}$ respectively. This parameter is subject to rather strong constraints if we want the correlation functions to have critical limits [21], and we also know which value of $\beta^2$ corresponds to a given weight $w$, among the infinitely many possibilities [3]. On the CFT side, the constraint on $\beta^2$ (58) is much weaker, and comes from the convergence of the operator product expansion.

In the case of non-contractible loops, the weight is related to a conformal dimension of a diagonal field $V_\Delta$ via the momentum $P$ (61). If the loop is around one vertex of valency zero, the diagonal field sits at that vertex, as already checked in the case of the sphere three-point functions via direct lattice calculations [9]. The loop weight is invariant under $P \to P + \beta^{-1}$, and we expect that the correct value of $P$ is the one that minimizes the real part of the conformal dimension. Bootstrap solutions for the other values of $P$ are also expected to contribute to the loop model correlation function, but only as subleading corrections that become negligible in the critical limit.

In the presence of a combinatorial map that is not weakly connected, there exist non-contractible loops that are not around one vertex. In this case, the corresponding diagonal field propagates in some channel, depending on the closed loop's combinatorial signature. In the case of sphere four-point functions, such diagonal fields appear in spectra of the type $\mathcal{S}^\Delta$ (66). The invariance of the weight under $P \to P + \beta^{-1}$ corresponds to the invariance of the corresponding interchiral block under the same shift.

With these relations between parameters, we expect:

**Conjecture 4.5** (Critical limit of correlation functions in loop models)
*When it exists, the critical limit of a loop model correlation function $Z_{M,W}$ (41) is a solution of the conformal bootstrap equations. Moreover, the set of correlation functions*

$$\left\{ \lim_{\text{critical}} Z_{M,W} \right\}_{M \in \mathcal{M}_{g,n}(r_1,\dots,r_n)}, \tag{76}$$

*is a basis of solutions of the corresponding conformal bootstrap equations.*

## 4.3 Numerical tests

We have tested our conjectures 4.2, 4.3 and 4.4 on the numbers of solutions of crossing symmetry equations, by numerically computing them in a number of cases. The principles of these numerical computations are explained in [3], and we do not repeat them here. Rather, we list the cases where tests were done.

### 4.3.1 Cases from previous work

The 30 simplest four-point functions of the $O(N)$ model were investigated in [3](Section 4.3). These four-point functions are of the type $\left\langle \prod_{i=1}^4 V_{(r_i,s_i)} \right\rangle$ with $r_i \in \frac{1}{2}\mathbb{N}^*$ and $\sum r_i \leq 4$. The spectrum is of course $\mathcal{S}^{O(N)}$ (64), or actually the subset of $\mathcal{S}^{O(N)}$ that is allowed by fusion rules, depending on the four-point function and on the channel. These results are consistent with Conjectures 4.2 and 4.3. Some cases are not immediately captured by the conjectures, due to the fusion rules of degenerate fields: the conjectures can then easily be adapted. For example, the four-point function $\left\langle V_{(1,0)} V_{(1,0)} V_{(1,1)} V_{(1,1)} \right\rangle$ has a 4-dimensional space of solutions whereas $\left| \mathcal{M}_{0,4}(1,1,1,1) \right| = 6$ and $\left| \mathcal{M}_{0,4}^c(1,1,1,1) \right| = 3$: in this case, fusion rules allow the

degenerate field to appear in the $s$-channel but not in the $t$- and $u$-channels, so that only one out of the three disconnected maps should be taken into account.

Four-point functions of diagonal fields $\left\langle \prod_{i=1}^{4} V_{\Delta_i} \right\rangle$ were investigated in detail in [5], with spectra of the type $\mathcal{S}^{\Delta}$. The space of solutions was found to be one-dimensional. This case was particularly useful for understanding the role of diagonal fields in the spectrum.

Four-point functions of the Potts model were investigated in [4](Section 4.3). More specifically, 28 four-point functions with $0 \leq \sum r_i \leq 6$ and $r_i \in \{0, 2, 3, 4\}$ were computed. Conjecture 4.3 predicts the numbers of solutions with the spectrum $\mathcal{S}^{\Delta_{(0, \frac{1}{2})}}$ in all channels, see Eq. (27a). However, to obtain the Potts model spectrum $\mathcal{S}^{\text{Potts}}$ (65), we have to remove the fields $V_{(1,0)}, V_{(1,1)}$, and to add degenerate fields whenever allowed by fusion rules. We now notice that for all cases with $3 \leq \sum r_i \leq 6$, the number of solutions from [4] agrees with Eq. (27a), minus 6, plus 1 whenever a degenerate field is allowed. This provides support for Conjecture 4.3, if we assume that removing the two primary fields $V_{(1,0)}, V_{(1,1)}$ in each one of the three channels reduces the number of solutions by 6.

For $\sum r_i \in \{0, 2\}$, things are more complicated. For $\left\langle V_{(2,s)} V_{\Delta_{(0, \frac{1}{2})}} V_{\Delta_{(0, \frac{1}{2})}} V_{\Delta_{(0, \frac{1}{2})}} \right\rangle$ with $s \in \{0, \frac{1}{2}, 1\}$, Eq. (27a) predicts 6 solutions with the spectrum $\mathcal{S}^{\Delta_{(0, \frac{1}{2})}}$. With the spectrum $\mathcal{S}^{\text{Potts}}$, 3 solutions are found. For $\left\langle V_{\Delta_{(0, \frac{1}{2})}} V_{\Delta_{(0, \frac{1}{2})}} V_{\Delta_{(0, \frac{1}{2})}} V_{\Delta_{(0, \frac{1}{2})}} \right\rangle$, we expect 1 solution with $\mathcal{S}^{\Delta_{(0, \frac{1}{2})}}$, and therefore again 1 solution with $\mathcal{S}^{\text{Potts}}$, after adding the degenerate fields $V_{\langle 1,2 \rangle}, V_{\langle 1,3 \rangle}$ but removing $V_{(1,0)}, V_{(1,1)}$. In fact, 4 solutions are observed. We conclude that in these cases, removing $V_{(1,0)}, V_{(1,1)}$ only kills 3 solutions, rather than 6. This is consistent with Conjecture 4.3, but only provides weak support, as long we do not know in which case removing a field from the spectrum actually reduces the number of solutions.

### 4.3.2 Systematic scan of examples

In Appendix A, we have listed the combinatorial maps with zero to three edges, and the connected maps with four or five edges, together with their signatures. Our list is complete modulo symmetries. Moreover, we have singled out the 53 maps such that $\left| \mathcal{M}_{0,4}(\delta_i | \sigma) \right| = 1$. For any such map, according to Conjecture 4.4, the corresponding space of solutions is one-dimensional.

With our numerical methods, one-dimensional spaces are particularly easy to handle. Actually, in order to determine the dimension of a space of solutions, we set a number of four-point structure constants to zero, until the space becomes one-dimensional. But there is always the risk of choosing a structure constant that is identically zero on the space in question, which would make us overestimate its dimension. This risk is absent if the space is one-dimensional. We can then single out one solution by normalizing a four-point structure constant to one, and the numerical results converge towards that solution when the numerical cutoffs increase.

For each combinatorial map, there are finitely many choices of $s_i \in \frac{1}{r_i}\mathbb{Z} \cap (-1, 1]$. We have not tested all the corresponding correlation functions. Rather, for each map such that $\left| \mathcal{M}_{0,4}(\delta_i | \sigma) \right| = 1$, we have tested the correlation function such that $s_i = 0$, plus another correlation function (unless there is no other, which happens in the three cases $(r_i) = (0, 0, 0, 0), (\frac{1}{2}, \frac{1}{2}, 0, 0), (\frac{1}{2}, \frac{1}{2}, \frac{1}{2}, \frac{1}{2}))$.

For example, let us consider the case $(r_i) = (\frac{5}{2}, 1, 1, \frac{1}{2})$. The 8 connected maps are displayed in Eq. (36). Modulo the $\mathbb{Z}_2$ symmetry that permutes the two bivalent vertices, we have the 4 maps of Eq. (A.28). Out of these 4 maps, the 2 maps with $\sigma = (\frac{5}{2}, 2, \frac{3}{2})$ and $\sigma = (\frac{1}{2}, 3, \frac{5}{2})$ obey $\left| \mathcal{M}_{0,4}(\delta_i | \sigma) \right| = 1$. In the second case, this means that the crossing symmetry equations with $\left( \mathcal{S}^{(s)}, \mathcal{S}^{(t)}, \mathcal{S}^{(u)} \right) = \left( \mathcal{S}_{\frac{1}{2}}, \mathcal{S}_3, \mathcal{S}_{\frac{5}{2}} \right)$ are conjectured to have a one-dimensional space of solutions. This applies to four-point functions with the stated values of $(r_i)$, starting

with $\left\langle V_{(\frac{5}{2},0)} V_{(1,0)} V_{(1,0)} V_{(\frac{1}{2},0)} \right\rangle$, but also including for example $\left\langle V_{(\frac{5}{2},\frac{4}{5})} V_{(1,1)} V_{(1,0)} V_{(\frac{1}{2},0)} \right\rangle$, or even $\left\langle V_{(\frac{5}{2},\frac{14}{5})} V_{(1,7)} V_{(1,-4)} V_{(\frac{1}{2},6)} \right\rangle$ — but we consider this latter four-point function equivalent to the previous one by the interchiral symmetry $s_i \rightarrow s_i + 2$.

In all tested cases, we have found a one-dimensional space of solutions, in agreement with Conjecture 4.4. (Our code is available at GitLab [22].) This suggests that our definition of the signature of a map is sound combinatorially, and relevant to conformal field theory. Moreover, most correlation functions belong neither to the $O(N)$ model, nor to the Potts model. This supports the idea that there exists a conformal field theory that includes and generalizes both models, and also includes diagonal fields with arbitrary conformal dimensions.

# 5 Concluding remarks

## 5.1 Solving loop models: the next steps

- If Conjecture 4.5 holds, we have a bijection between combinatorial maps and solutions of conformal bootstrap equations, and the obvious question is: which map corresponds to which solution? We know it in quite a few cases, which are listed in bold in Appendix A. However, we do not know it in general. A solution can in principle be singled out by imposing a number of linear constraints in addition to the conformal bootstrap equations. Constraints may include the vanishing of some four-point structure constants, or more general linear relations.

- After solving crossing symmetry as a linear system of equations for four-point structure constants, it remains to factorize four-point structure constants into three-point structure constants as in Eq. (56). In the case of correlation functions of diagonal fields, factorization has been investigated numerically, but it is not clear how to interpret the results [5].

- Understanding factorization would be essential for defining fusion in loop models [23, 24], and more generally for interpreting correlation functions in the context of a field theory with well-defined operator product expansions. Do our correlation functions fit in standard axiomatic formalisms of two-dimensional CFT such as the Moore–Seiberg formalism [25] or the Fuchs–Runkel–Schweigert formalism [26]? The latter formalism relies on topological objects whereas our maps are combinatorial, presumably because that formalism accomodates conformal blocks whereas we are only dealing with single-valued correlation functions.

## 5.2 More evidence for the conjectures, please!

We are ready to admit that our numerical bootstrap results are not as far-reaching as our conjectures. More tests of the conjectures would be welcome. Tests that could be performed using existing techniques include:

- Numerically solving conformal bootstrap equations for higher correlation functions $n > 4$ and/or in higher genus $g > 0$. Our results are limited to four-point functions on the sphere $(g, n) = (0, 4)$, because this is the first nontrivial case, with correlation functions that depend on one geometric modulus, namely the cross-ratio of the four positions. The number of geometric moduli for an $n$-point function in genus $g$ is $3g - 3 + n$. There is another case with one modulus: the one-point function on the torus; however, the conformal maps in this case are rather trivial. Cases with more moduli would certainly be

numerically challenging, and no longer accessible by running Python code on standard computers for a few minutes.

- Directly testing Conjecture 4.5 on the critical limit of correlation functions is doable in principle. Such correlation functions can be computed on the lattice by transfer matrix or Monte-Carlo methods. The problem is that rather large lattices would be needed for reaching a good precision, especially if we wanted to accurately compute angles.

### 5.3 Generalizations: A wish list

- We do understand the spaces of solutions that are relevant to the $O(N)$ model [3], but the same cannot be said of the Potts model. The Potts model does not have the fields $V_{(1,0)}$ and $V_{(1,1)}$, and we do not know in general how their elimination affects the number of solutions. Global symmetry can help us single out the relevant solutions in simple cases [4], but not in general. On the side of combinatorial maps, the lattice definition of the model implies that maps are bicolorable, with all vertices of valency zero on faces of the same color [27]. Bicolorability just implies $r \in \mathbb{N}$, and the further constraint eliminates some maps in some cases, without solving the problem.

- The case of Riemann surfaces with boundaries would be interesting. Our approach is able to account for the various primary fields of loop models via the weights of loop configurations: could it also account for all conformal boundary conditions? The principles of conformal invariance and single-valuedness, which led us to conjecturally solve crossing symmetry in terms of combinatorial maps, might also determine which boundary conditions are possible. In particular, it would be nice to understand whether there is a bijection between boundary conditions and primary fields, as in theories that are diagonal and rational [28].

- A challenge would be to understand the Potts and $O(N)$ models in higher dimensions, and in particular the numbers of solutions of crossing symmetry in these models. In their spectra, there are probably fewer degeneracies than in two dimensions (if any), which makes the problem simpler. And we expect that correlation functions are well described by global symmetry invariants, according to the following heuristic argument: invariants are obtained by contracting tensor indices, and the possible contractions can be represented as graphs. In two dimensions, bootstrap solutions correspond to planar graphs, and this is why we can have fewer solutions than invariants. In higher dimensions, without the constraint of planarity, there should be as many solutions as invariants.

### 5.4 Which crossing symmetry equations do we want to solve?

We have been focussing on crossing symmetry equations under the assumption of interchiral symmetry. By our definition, interchiral symmetry determines how structure constants behave under $s \to s + 2$ where $s$ is the second Kac index, as follows from the existence of the degenerate field $V_{(1,3)}$. However, in some Potts model correlation functions such as connectivities, interchiral symmetry is enhanced to $s \to s + 1$, because all relevant three-point structure constants involve the field $V_{\Delta_{(0,\frac{1}{2})}}$ [18]. Conversely, we could relax interchiral symmetry, and work with the original conformal blocks, rather than interchiral blocks.

In the example of the four-point function of $V_{\Delta_{(0,\frac{1}{2})}}$, we have numerically found that tightening interchiral symmetry can indeed lower the number of crossing symmetry solutions, while relaxing interchiral symmetry can increase it. It would be particularly interesting to understand the extra solutions that violate interchiral symmetry, and whether they belong to a CFT

without any degenerate field. Let us display the numbers of solutions that we find, depending on the spectrum (the same in all channels) and on the interchiral symmetry:

| Spectrum | $s \to s+2$ | $s \to s+1$ |
|:---:|:---:|:---:|
| $\mathcal{S}^{\text{Potts}} \cup \mathcal{S}^{O(N)}$ | 7 | 4 |
| $\mathcal{S}^{\text{Potts}}$ | 4 | 4 |
| $\mathcal{S}^{\Delta_{(0,\frac{1}{2})}}$ | 1 | 1 |
| $\mathcal{S}^{\Delta_{(0,s)}} \cup \mathcal{S}^{\Delta_{(0,s+1)}}$ | 4 | 1 |

(77)

For generic values of $\Delta$, the enhanced interchiral symmetry makes no sense for the spectrum $\mathcal{S}^{\Delta}$ if $\Delta \neq \Delta_{(0,\frac{1}{2})}$. We therefore introduce the spectrum $\mathcal{S}^{\Delta_{(0,s)}} \cup \mathcal{S}^{\Delta_{(0,s+1)}}$, which is invariant under $s \to s+1$, so that it makes sense to impose the enhanced interchiral symmetry on structure constants.

## 5.5 Why these maps, why these weights, why these loop ensembles?

We have defined combinatorial maps, and the weights in sums over loop configurations, in order to reproduce CFT correlation functions with all their parameters, and to recover numerical results on numbers of solutions of crossing symmetry. It would be interesting to have more intrinsic justifications for these definitions, and to explore possible generalizations. Questions include:

- Why do we have to forbid monogons? This basic assumption is convenient combinatorially, as it eliminates many maps, so that numbers of maps depend polynomially on the valencies, rather than factorially. It is also convenient for computing angles on the lattice, since our proposed scheme (51) would not work in the presence of small monogons. And monogons can be consistently eliminated in the lattice model, using projectors of the Jones–Wenzl type in the corresponding diagram algebras. However, all this only shows that it is easy to forbid monogons, without providing a compelling reason for doing so.

  In the related problem of polymer networks, monogons lead to divergences that are dealt with by renormalization [29,30]. The idea is that in the critical limit, a vertex of valency $2r$ with two half-edges that form a monogon actually behaves like a vertex of valency $2r-2$. This behaviour is not specific to two dimensions, and provides a physical reason for ignoring monogons.

- Why do all closed loops have the same weight? Given a combinatorial map with marked vertices, we can usually distinguish the faces, and the weight of a closed loop could depend on the face it lives in. The CFT interpretation of the resulting correlation function would be tricky, because it would be map-dependent, and involve different values of the central charge (which is a function of the weight of closed loops). A possibility would be that the combinatorial map determines the positions of defects that change the central charge.

- Instead of or in addition to loops, we could consider other variables that give rise to equivalent representations of the same statistical models: spins, Fortuin–Kasteleyn clusters, flows. This could result in different spectra of primary fields, and different combinatorial representations of correlation functions.

## Acknowledgements

We are grateful to Jérémie Bouttier, Séverin Charbonnier, Bertrand Duplantier, Emmanuel Guitter, Paul Norbury, and Paul Roux, for valuable discussions and correspondence. Moreover, we wish to thank Jérémie Bouttier, Emmanuel Guitter, Adam Nahum and Paul Roux for helpful suggestions on the draft text. We are grateful to Bernard Nienhuis for the review he wrote for SciPost, which led to significant clarifications.

Rongvoram Nivesvivat gratefully acknowledges the personal hospitality of the Jia family in Beijing in Fall 2022.

**Funding information**  This work is partly a result of the project ReNewQuantum, which received funding from the European Research Council. This work was also supported by the French Agence Nationale de la Recherche (ANR) under grant ANR-21-CE40-0003 (project CONFICA).

## A  List of examples

In this appendix we systematically display planar maps by increasing number of edges $\sum r_i \in \mathbb{N}$. We display all maps with $\sum r_i \leq 3$, and all connected maps with $\sum r_i \leq 5$.

To save space and avoid redundancies, we sometimes make use of the symmetry of our sets of maps under permutations of vertices and/or edges. Such symmetries exist whenever two or more vertices have the same valency. In such cases, a displayed map may come with a multiplicity $m$, which is the size of a relevant group of permutations. (We do not necessarily use the largest available group.) Then $m - 1$ is the number of similar maps that we do not display explicitly.

For each map, we indicate the signature $\sigma$, see Definition 2.6. Two maps that are related by a symmetry may or may not have the same signature. The signature is written in **bold** if our set of maps $\mathcal{M}_{0,4}$ contains no other map with a larger or equal signature $\sigma' \geq \sigma$ — in other words, if $\left| \mathcal{M}_{0,4}(\delta_i | \sigma) \right| = 1$. In this case, Conjecture 4.4 states that the corresponding crossing symmetry equations have a one-dimensional space of solutions.

### A.1  All planar maps with zero to three edges

#### A.1.1  Case $\left| \mathcal{M}_{0,4}(0,0,0,0) \right| = 1$

$$
\begin{matrix}
\times & & \times \\
& & \\
\times & & \times
\end{matrix}
\tag{A.1}
$$

$$\sigma = (0,0,0)$$

#### A.1.2  Case $\left| \mathcal{M}_{0,4}(\frac{1}{2}, \frac{1}{2}, 0, 0) \right| = 1$

$$
\begin{matrix}
\bullet & & \times \\
| & & \\
\bullet & & \times
\end{matrix}
\tag{A.2}
$$

$$\sigma = (0, \tfrac{1}{2}, \tfrac{1}{2})$$

### A.1.3 Case $\left|\mathcal{M}_{0,4}(1,0,0,0)\right| = 3$

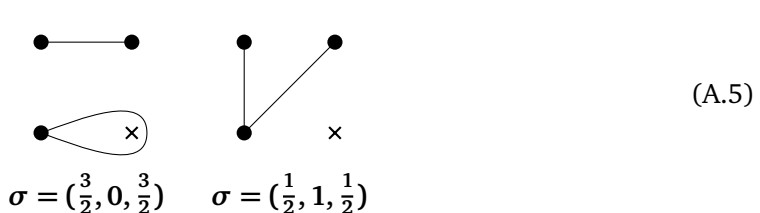

$$\sigma = (0,1,1)$$
$$m = 3$$

(A.3)

### A.1.4 Case $\left|\mathcal{M}_{0,4}(\frac{1}{2},\frac{1}{2},\frac{1}{2},\frac{1}{2})\right| = 3$

$$\sigma = (0,1,1)$$
$$m = 3$$

(A.4)

### A.1.5 Case $\left|\mathcal{M}_{0,4}(1,\frac{1}{2},\frac{1}{2},0)\right| = 2$

(A.5)

$$\sigma = (\tfrac{3}{2},0,\tfrac{3}{2}) \qquad \sigma = (\tfrac{1}{2},1,\tfrac{1}{2})$$

### A.1.6 Case $\left|\mathcal{M}_{0,4}(1,1,0,0)\right| = 4$

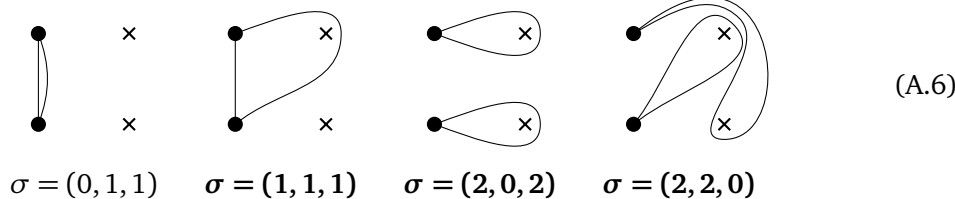

(A.6)

$$\sigma = (0,1,1) \qquad \sigma = (1,1,1) \qquad \sigma = (2,0,2) \qquad \sigma = (2,2,0)$$

### A.1.7 Case $\left|\mathcal{M}_{0,4}(\frac{3}{2},\frac{1}{2},0,0)\right| = 3$

(A.7)

$$\sigma = (0,\tfrac{3}{2},\tfrac{3}{2}) \qquad \sigma = (1,\tfrac{1}{2},\tfrac{3}{2})$$
$$m = 2$$

### A.1.8   Case $\left|\mathcal{M}_{0,4}(2,0,0,0)\right| = 6$

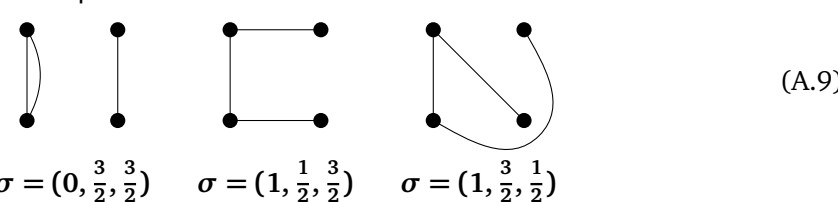

(A.8)

$\sigma = (0,2,2)$   $\sigma = (1,1,2)$
$m = 3$          $m = 3$

### A.1.9   Case $\left|\mathcal{M}_{0,4}(1,1,\frac{1}{2},\frac{1}{2})\right| = 3$

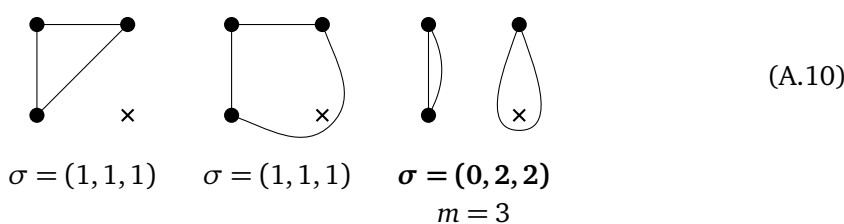

(A.9)

$\sigma = (0,\frac{3}{2},\frac{3}{2})$   $\sigma = (1,\frac{1}{2},\frac{3}{2})$   $\sigma = (1,\frac{3}{2},\frac{1}{2})$

### A.1.10   Case $\left|\mathcal{M}_{0,4}(1,1,1,0)\right| = 5$

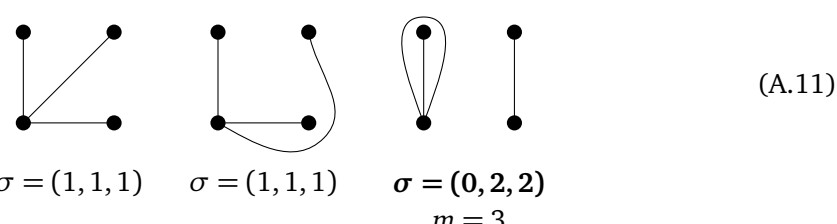

(A.10)

$\sigma = (1,1,1)$   $\sigma = (1,1,1)$   $\sigma = (0,2,2)$
$m = 3$

### A.1.11   Case $\left|\mathcal{M}_{0,4}(\frac{3}{2},\frac{1}{2},\frac{1}{2},\frac{1}{2})\right| = 5$

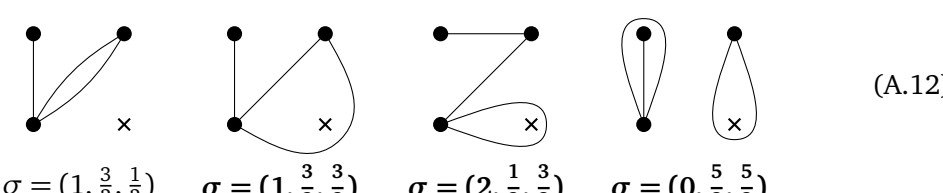

(A.11)

$\sigma = (1,1,1)$   $\sigma = (1,1,1)$   $\sigma = (0,2,2)$
$m = 3$

### A.1.12   Case $\left|\mathcal{M}_{0,4}(\frac{3}{2},\frac{1}{2},1,0)\right| = 4$

(A.12)

$\sigma = (1,\frac{3}{2},\frac{1}{2})$   $\sigma = (1,\frac{3}{2},\frac{3}{2})$   $\sigma = (2,\frac{1}{2},\frac{3}{2})$   $\sigma = (0,\frac{5}{2},\frac{5}{2})$

### A.1.13 Case $\left|\mathcal{M}_{0,4}(\tfrac{3}{2},\tfrac{3}{2},0,0)\right| = 5$

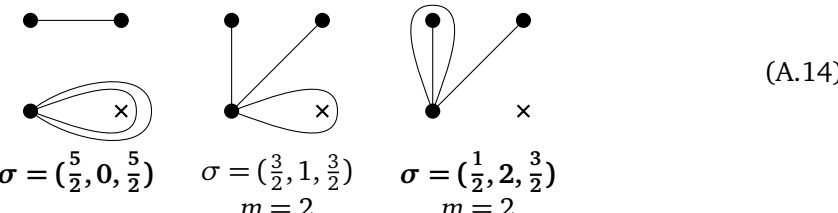

$$\sigma = (0,\tfrac{3}{2},\tfrac{3}{2}) \qquad \sigma = (1,\tfrac{3}{2},\tfrac{3}{2}) \qquad \sigma = (2,\tfrac{1}{2},\tfrac{5}{2})$$
$$m=2 \qquad\qquad m=2$$

(A.13)

### A.1.14 Case $\left|\mathcal{M}_{0,4}(2,\tfrac{1}{2},\tfrac{1}{2},0)\right| = 5$

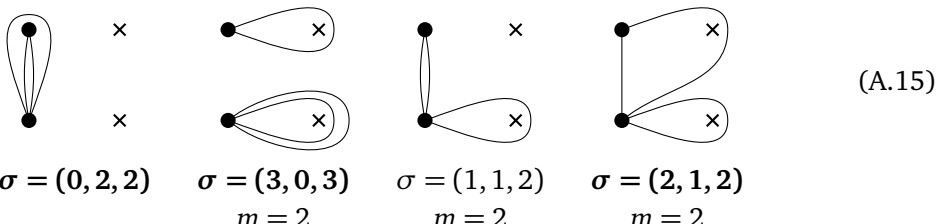

$$\sigma = (\tfrac{5}{2},0,\tfrac{5}{2}) \qquad \sigma = (\tfrac{3}{2},1,\tfrac{3}{2}) \qquad \sigma = (\tfrac{1}{2},2,\tfrac{3}{2})$$
$$m=2 \qquad\qquad m=2$$

(A.14)

### A.1.15 Case $\left|\mathcal{M}_{0,4}(2,1,0,0)\right| = 7$

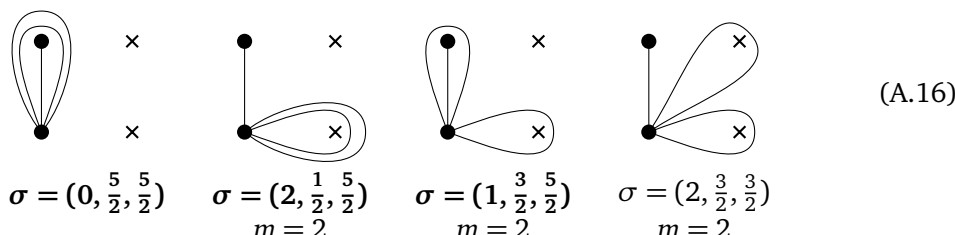

$$\sigma = (0,2,2) \qquad \sigma = (3,0,3) \qquad \sigma = (1,1,2) \qquad \sigma = (2,1,2)$$
$$m=2 \qquad\quad m=2 \qquad\quad m=2$$

(A.15)

### A.1.16 Case $\left|\mathcal{M}_{0,4}(\tfrac{5}{2},\tfrac{1}{2},0,0)\right| = 7$

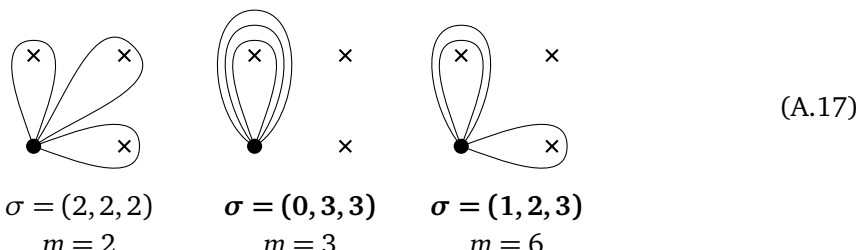

$$\sigma = (0,\tfrac{5}{2},\tfrac{5}{2}) \qquad \sigma = (2,\tfrac{1}{2},\tfrac{5}{2}) \qquad \sigma = (1,\tfrac{3}{2},\tfrac{5}{2}) \qquad \sigma = (2,\tfrac{3}{2},\tfrac{3}{2})$$
$$m=2 \qquad\qquad m=2 \qquad\qquad m=2$$

(A.16)

### A.1.17 Case $\left|\mathcal{M}_{0,4}(3,0,0,0)\right| = 11$

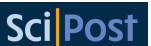

$$\sigma = (2,2,2) \qquad \sigma = (0,3,3) \qquad \sigma = (1,2,3)$$
$$m=2 \qquad\qquad m=3 \qquad\qquad m=6$$

(A.17)

## A.2  All connected planar maps with four or five edges

### A.2.1  Case $\left|\mathcal{M}_{0,4}^{c}(1,1,1,1)\right| = 3$

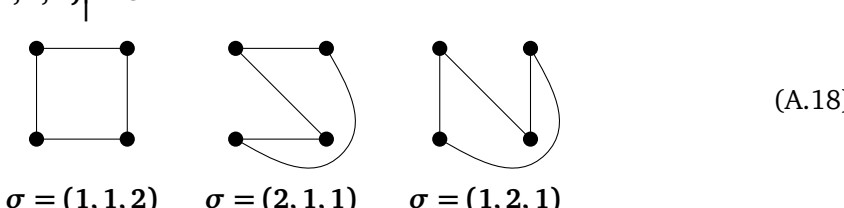

$$\sigma = (1,1,2) \qquad \sigma = (2,1,1) \qquad \sigma = (1,2,1) \tag{A.18}$$

### A.2.2  Case $\left|\mathcal{M}_{0,4}^{c}(\frac{3}{2},1,1,\frac{1}{2})\right| = 4$

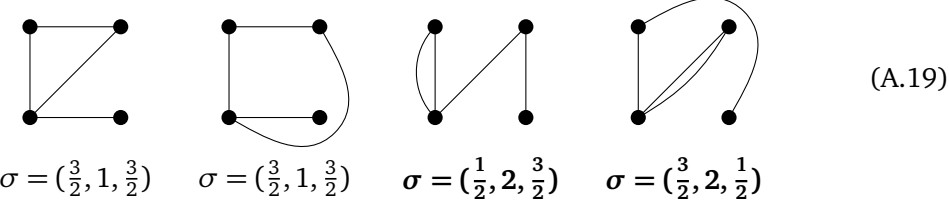

$$\sigma = (\tfrac{3}{2},1,\tfrac{3}{2}) \qquad \sigma = (\tfrac{3}{2},1,\tfrac{3}{2}) \qquad \sigma = (\tfrac{1}{2},2,\tfrac{3}{2}) \qquad \sigma = (\tfrac{3}{2},2,\tfrac{1}{2}) \tag{A.19}$$

### A.2.3  Case $\left|\mathcal{M}_{0,4}^{c}(\frac{3}{2},\frac{3}{2},\frac{1}{2},\frac{1}{2})\right| = 4$

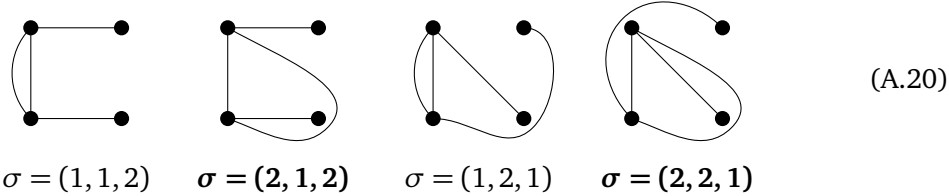

$$\sigma = (1,1,2) \qquad \sigma = (2,1,2) \qquad \sigma = (1,2,1) \qquad \sigma = (2,2,1) \tag{A.20}$$

### A.2.4  Case $\left|\mathcal{M}_{0,4}^{c}(2,1,\frac{1}{2},\frac{1}{2})\right| = 5$

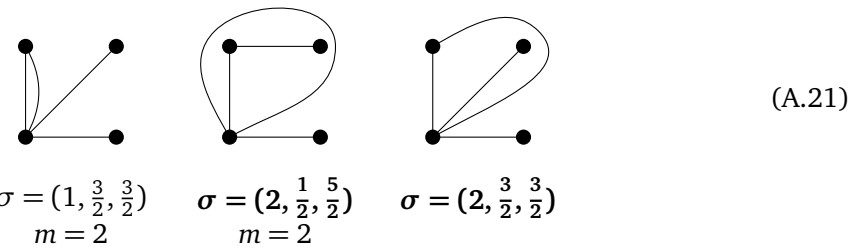

$$\sigma = (1,\tfrac{3}{2},\tfrac{3}{2}) \qquad \sigma = (2,\tfrac{1}{2},\tfrac{5}{2}) \qquad \sigma = (2,\tfrac{3}{2},\tfrac{3}{2}) \tag{A.21}$$
$$m = 2 \qquad\qquad m = 2$$

### A.2.5  Case $\left|\mathcal{M}_{0,4}^{c}(\frac{5}{2},\frac{1}{2},\frac{1}{2},\frac{1}{2})\right| = 6$

$$\sigma = (2,1,2) \qquad \sigma = (2,1,2) \tag{A.22}$$
$$m = 3 \qquad\qquad m = 3$$

### A.2.6 Case $\left|\mathcal{M}_{0,4}^c(\frac{3}{2},\frac{3}{2},1,1)\right| = 6$

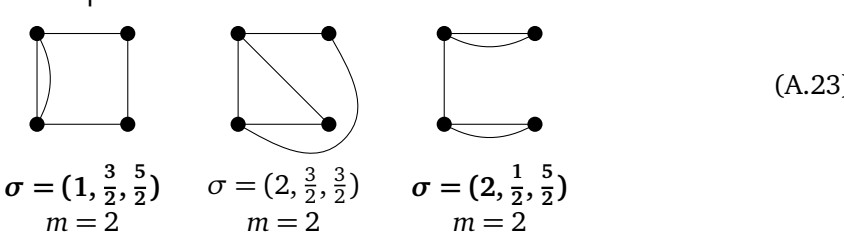

$$\text{(A.23)}$$

$\sigma = (1,\frac{3}{2},\frac{5}{2})$ $\qquad$ $\sigma = (2,\frac{3}{2},\frac{3}{2})$ $\qquad$ $\sigma = (2,\frac{1}{2},\frac{5}{2})$
$m = 2$ $\qquad\qquad$ $m = 2$ $\qquad\qquad$ $m = 2$

### A.2.7 Case $\left|\mathcal{M}_{0,4}^c(\frac{3}{2},\frac{3}{2},\frac{3}{2},\frac{1}{2})\right| = 6$

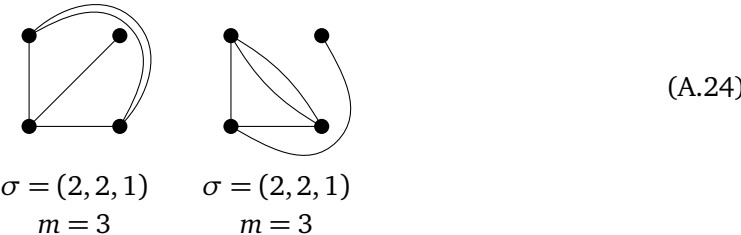

$$\text{(A.24)}$$

$\sigma = (2,2,1)$ $\qquad$ $\sigma = (2,2,1)$
$m = 3$ $\qquad\qquad$ $m = 3$

### A.2.8 Case $\left|\mathcal{M}_{0,4}^c(2,1,1,1)\right| = 6$

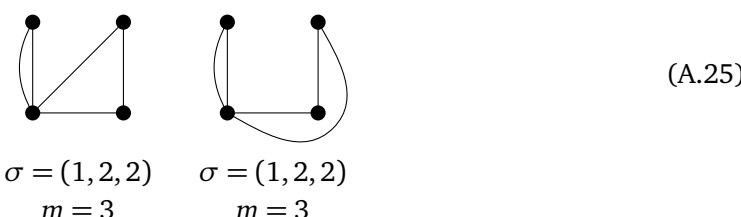

$$\text{(A.25)}$$

$\sigma = (1,2,2)$ $\qquad$ $\sigma = (1,2,2)$
$m = 3$ $\qquad\qquad$ $m = 3$

### A.2.9 Case $\left|\mathcal{M}_{0,4}^c(2,\frac{3}{2},\frac{1}{2},1)\right| = 7$

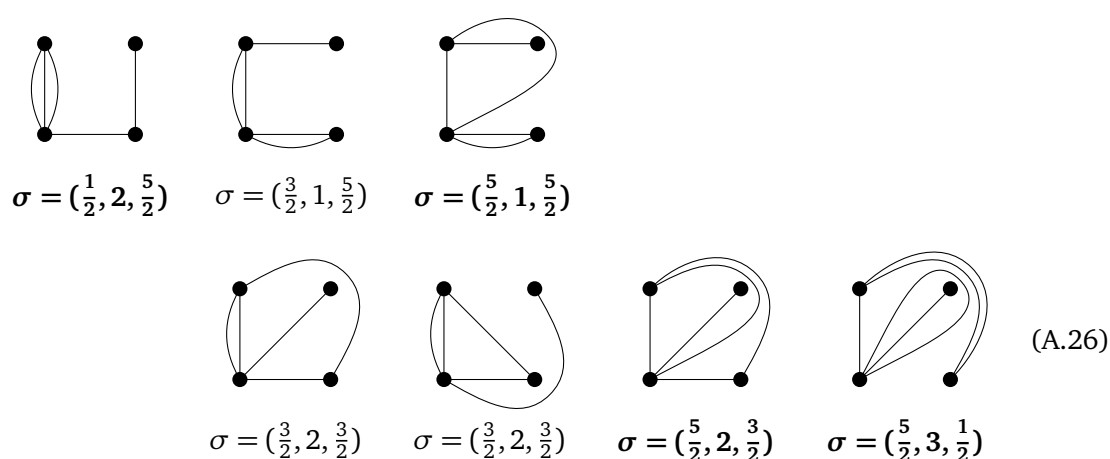

$\sigma = (\frac{1}{2},2,\frac{5}{2})$ $\quad$ $\sigma = (\frac{3}{2},1,\frac{5}{2})$ $\quad$ $\sigma = (\frac{5}{2},1,\frac{5}{2})$

$$\text{(A.26)}$$

$\sigma = (\frac{3}{2},2,\frac{3}{2})$ $\quad$ $\sigma = (\frac{3}{2},2,\frac{3}{2})$ $\quad$ $\sigma = (\frac{5}{2},2,\frac{3}{2})$ $\quad$ $\sigma = (\frac{5}{2},3,\frac{1}{2})$

**A.2.10**  Case $\left|\mathcal{M}_{0,4}^c(2,2,\frac{1}{2},\frac{1}{2})\right| = 8$

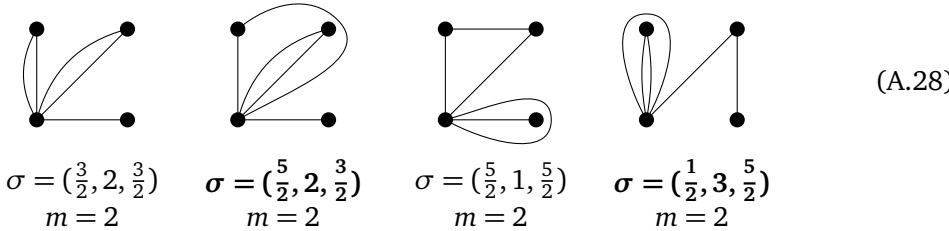

$$\begin{array}{cccc}
\sigma = (1,\frac{3}{2},\frac{5}{2}) & \sigma = (2,\frac{3}{2},\frac{5}{2}) & \sigma = (2,\frac{3}{2},\frac{5}{2}) & \boldsymbol{\sigma = (3,\frac{1}{2},\frac{7}{2})} \\
m = 2 & m = 2 & m = 2 & m = 2
\end{array}$$

(A.27)

**A.2.11**  Case $\left|\mathcal{M}_{0,4}^c(\frac{5}{2},1,1,\frac{1}{2})\right| = 8$

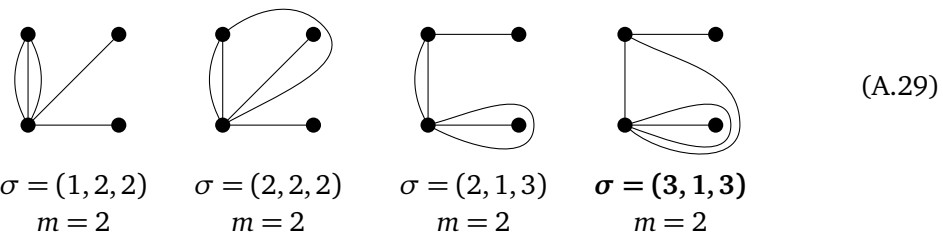

$$\begin{array}{cccc}
\sigma = (\frac{3}{2},2,\frac{3}{2}) & \boldsymbol{\sigma = (\frac{5}{2},2,\frac{3}{2})} & \sigma = (\frac{5}{2},1,\frac{5}{2}) & \boldsymbol{\sigma = (\frac{1}{2},3,\frac{5}{2})} \\
m = 2 & m = 2 & m = 2 & m = 2
\end{array}$$

(A.28)

**A.2.12**  Case $\left|\mathcal{M}_{0,4}^c(\frac{5}{2},\frac{3}{2},\frac{1}{2},\frac{1}{2})\right| = 8$

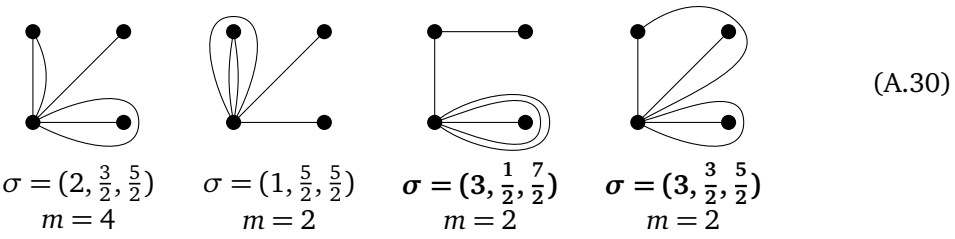

$$\begin{array}{cccc}
\sigma = (1,2,2) & \sigma = (2,2,2) & \sigma = (2,1,3) & \boldsymbol{\sigma = (3,1,3)} \\
m = 2 & m = 2 & m = 2 & m = 2
\end{array}$$

(A.29)

**A.2.13**  Case $\left|\mathcal{M}_{0,4}^c(3,1,\frac{1}{2},\frac{1}{2})\right| = 10$

$$\begin{array}{cccc}
\sigma = (2,\frac{3}{2},\frac{5}{2}) & \sigma = (1,\frac{5}{2},\frac{5}{2}) & \boldsymbol{\sigma = (3,\frac{1}{2},\frac{7}{2})} & \boldsymbol{\sigma = (3,\frac{3}{2},\frac{5}{2})} \\
m = 4 & m = 2 & m = 2 & m = 2
\end{array}$$

(A.30)

**A.2.14**  Case $\left|\mathcal{M}_{0,4}^c(\frac{7}{2},\frac{1}{2},\frac{1}{2},\frac{1}{2})\right| = 12$

$$\begin{array}{cc}
\sigma = (2,2,3) & \sigma = (3,1,3) \\
m = 6 & m = 6
\end{array}$$

(A.31)

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
