# Peer review of "From combinatorial maps to correlation functions in loop models"

_SciPost Physics, doi:SciPost Phys. 15, 147 (2023)_

## Round 1 · Referee Report · Anonymous (Referee 1) · 2023-6-15

Report
The O(N) and Potts models are generalizations of the Ising model which play an important role in the theory of critical phenomena. It is known that in two dimensions the Coulomb gas method leads to formulations of these models in terms of nonintersecting loops, and that in these formulations the global symmetries characteristic of the two models - O(N) and permutational, respectively - are no longer easy to identify. The authors of the present paper observe that the role of global symmetry may be similarly difficult to identify when looking for solutions of the crossing symmetry equations within the two-dimensional conformal bootstrap for correlation functions at criticality. The idea of the paper is then to propose a framework in which suitably defined correlation functions may provide a basis for solutions of the conformal bootstrap equations; this is pursued without reference to global symmetry, and then without a specific focus on O(N) and Potts models.
The programme is ambitious and several steps are admittedly conjectural. The authors associate a correlation function to a combinatorial map, namely a graph made of vertices and edges, further constrained by a number of requirements designed to make contact with the properties of two-dimensional conformal field theory. In particular, the number 2r of edges ending at a vertex and the angular momentum s depending on the angles of the edges at the vertex are associated to the indices r and s labeling conformal dimensions within the Kac parametrization. A correspondence of similar nature is provided on the lattice by the Coulomb gas approach, but the authors hope that their combinatorial definitions make sense of it directly in the continuum. As they recall, it is known that the connectivities of the random cluster model can be mapped onto correlation functions of local fields directly in the continuum.
The arguments developed by the authors lead them to propose identifications between dimensions of spaces of solutions of crossing symmetry equations in theories with specific field contents and numbers of combinatorial maps belonging to suitably chosen sets. Then they use the numerical bootstrap for four-point correlation functions to verify that the conjectures are indeed confirmed in a number of cases they can test. They make explicit the conjectural nature of the paper and point out that the numerical verifications they present are limited and that more will be needed.
The paper provides an interesting step in the investigation of a vast and difficult problem. I suggest publication in SciPost.

---

## Round 1 · Referee Report · Bernard Nienhuis (Referee 2) · 2023-7-27

Strengths
1) The definition of the combinatorial maps is very clear and systematic, and helpfully enlightened with examples.
2) The results are are significant progress in the understanding of correlation functions in loop models.
Weaknesses
1) While the approach is clearly stated, it is not clear where it comes from. The role of combinatorial maps for the definition of correlation functions seems to appear out of nowhere.
Report
Much information about the universal properties of the ordering phase transitions in so-called Potts and O(N) models in two dimensions has been obtained by the study of loop models, of which the states are configurations of loops in the plane. These loop configurations are intrinsically invariant under the global symmetry groups, the permutation group of Q elements and the rotation group in N dimensions respectively. As a result the action and representations of these symmetry groups are not at all obvious in terms of these loop models.
The authors of the present paper together with several others make an effort to clarify this problem in a series of recent papers: the current paper and its references [1], [3], [4] and [5].
This paper addresses the problem of correlation functions. It sets up a intricate way of defining different correlation functions based on combinatorial maps. Each (weakly connected) combinatorial map corresponds to one unique correlation function, i.e. one solution to the bootstrap equations. About the nature of the correlation function the authors say "it is not clear how to interpret it as a correlation function of local fields". It sounds as if it is a complete mystery. But it seems the authors know much more. I am unable to interpret the text in terms of the relation between the conbinatorial map and the correlation function. When the segments are discussed and the meaning of the angle with which they are incident to a vertex they seem to be related to the height in the height representation of the loop model. But what the relation is, is not made entirely explicit. The relation between the combinatorial map and the Kac classification of the correlated fields does appear to be clear.
While the combinatorial maps are a discrete set, taking into account the angles at which the segments join the vertices, appears to a be continuous defining parameters. it remains unclear to me if this is to be interpreted as a continuous family of correlation functions.
It is helpful that the authors are explicit in the aspects they do not yet know, but wish to know.
In spite of the fact that important aspects of the paper escape me, I do consider it an important and impressive project. In my opinion it deserves to be published in scipost. But I do hope that the unclarites mentioned above can be either resolved or stated explicitly.

---

## Round 2 · Author Response

We have made clarifications on the points that were raised in Bernard Nienhuis' report.

---

## Round 2 · List of Changes

- On page 5, in order to clarify the origin of combinatorial maps, we have rewritten the first full paragraph after Figure (1.4). ("The main idea...")

- On page 5, we have rewritten the last paragraph before the Highlights, in order to improve the discussion of the interpretation of correlation functions in terms of local fields.

- On page 18, we have enumerated the parameters of our sum over loop configurations, in order to clarify which parameters are discrete or continuous. In particular, the angles are parameters of the loops themselves and not of the sum. The angular momentums are discrete, and the only continuous parameters of the sum are the weights of closed loops.

- On page 21, we have rewritten the last paragraph of Section 3 in order to better conclude the comparison with the Coulomb gas approach.

You are currently on this page

Resubmission 2302.08168v2 on 25 August 2023

---

## Editorial Decision

published